# Bayesian Gated Non-Negative Contrastive Learning

Peng Cui [* 1]   Jiahao Zhang [* 1]   Lijie Hu [1]

## Abstract

While Contrastive Learning (CL) has revolutionized self-supervised representation learning, its latent representations remain highly entangled and opaque, limiting their interpretability in safety-critical applications. We identify that a fundamental cause of this entanglement is the reliance on deterministic similarity measures, which treat all feature dimensions equally. In compositional scenes, this creates an Optimization Conflict: common background features, such as, "blue sky", are encouraged to align in positive pairs but simultaneously repelled in negative pairs, causing gradient oscillations that hinder precise semantic disentanglement. To address this, we propose **BayesNCL** (Bayesian Gated Non-Negative Contrastive Learning). Unlike standard approaches, BayesNCL introduces a probabilistic gating mechanism that dynamically filters out task-irrelevant, high-frequency common features while selectively retaining discriminative semantics. By formalizing feature selection as a variational inference problem with a sparse Bernoulli prior, our method effectively resolves the optimization conflict. Empirical experimental results on Imagenet-100 demonstrate that BayesNCL achieves a remarkable 142.1% improvement in semantic consistency compared to state-of-the-art baselines, yielding highly interpretable representations without compromising downstream task performance. Code is available at https://github.com/Cui-Peng-624/BayesNCL.

## 1. Introduction

In recent years, self-supervised learning (SSL) (Jing & Tian, 2020), especially contrastive learning (CL) (Jaiswal et al., 2020), has emerged as a powerful paradigm for learning

---
[*]Equal contribution [1]Mohamed bin Zayed University of Artificial Intelligence (MBZUAI). Correspondence to: Lijie Hu <lijie.hu@mbzuai.ac.ae>.

*Proceedings of the $43^{rd}$ International Conference on Machine Learning*, Seoul, South Korea. PMLR 306, 2026. Copyright 2026 by the author(s).

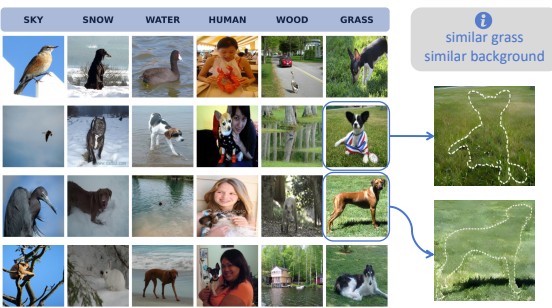

*Figure 1.* **The Compositionality Gap.** Real-world image features are combined. For example, we found that on Imagenet-100, many images have similar background features.

generalizable visual representations from unlabeled data. By optimizing instance discrimination objectives that pull augmented views of the same image together while pushing distinct images apart, CL effectively encodes high-level semantic information into compact feature vectors and demonstrates strong transferability across downstream tasks (Wu et al., 2018; Chen et al., 2020; He et al., 2020; Ericsson et al., 2021). Technically, these methods typically map inputs into a continuous, real-valued latent space, where feature dimensions are dense and allow for arbitrary rotation to maximize separability on a hypersphere (Wang & Isola, 2020).

Although contrastive learning has shown strong potential through its competitive performance, the standard contrastive paradigm still faces several hurdles in practice. Early challenges, such as the reliance on massive negative pairs and the risk of dimensional collapse, have been extensively explored and largely mitigated through innovations like memory banks (He et al., 2020), negative-free architectures (Grill et al., 2020; Tian et al., 2021), and feature decorrelation objectives (Zbontar et al., 2021; Bardes et al., 2021). However, a more fundamental limitation remains persistent and under-addressed: standard CL representations suffer from a severe lack of interpretability. In typical contrastive frameworks, the learned embedding space is highly entangled, where semantic concepts are distributed holistically across all feature dimensions rather than being aligned with specific axes (Wang et al., 2024). Consequently, individual dimensions lack specific semantic meaning (polysemy), rendering state-of-the-art models as opaque "black

boxes" (Chen et al., 2026; Lin et al., 2026; Lai et al., 2024; Hu et al., 2024a; 2025). This lack of interpretability poses significant risks for safety-critical fields (Thirunavukarasu et al., 2023; Li et al., 2023; Hu et al., 2024b; Zhou et al., 2025; Yang et al., 2026; 2025; Yao et al., 2025; Li et al., 2025; Zhang et al., 2025), where verifying the decision-making process of a model is as crucial as its predictive accuracy.

To specifically address this problem, Wang et al. (2024) recently proposed Non-Negative Contrastive Learning (NCL). By enforcing non-negativity, NCL establishes a theoretical equivalence to Non-negative Matrix Factorization (NMF) (Lee & Seung, 1999), successfully aligning feature dimensions with semantic clusters. Nevertheless, while NCL significantly improves interpretability, we identify a critical **Optimization Conflict** inherent in its deterministic nature. As illustrated in Figure 1 and Figure 2, real-world data is always compositional, distinct objects often share common high-frequency features, for example, a "blue sky" background. Consider a negative pair consisting of a bird and a plane against the same sky: the contrastive loss attempts to repel these images, forcing the model to suppress the shared "sky" dimension. Conversely, a positive pair, such as two views of the bird, requires aligning this same dimension. This optimization conflict induces severe gradient oscillation on shared dimensions, effectively obstructing the emergence of the clean, disentangled representations that NCL inherently seeks to achieve, we will elaborate on this in section 4.

To tackle this challenge, we argue that the model requires a mechanism to dynamically suppress these ambiguous common features during contrastive alignment. By probabilistically filtering out such task-irrelevant information, the optimization process can focus exclusively on discriminative semantic components, thereby resolving the gradient conflict. Driven by this insight, we propose Bayesian gated Non-Negative Contrastive Learning (**BayesNCL**). Instead of deterministic activation, we introduce a Bayesian Gating Head that learns a Bernoulli distribution for each feature dimension, allowing the model to adaptively "switch off" conflict-inducing dimensions. Empirically, this modification yields a substantial leap in feature interpretability: BayesNCL achieves a remarkable 142.1% relative improvement in semantic consistency compared to the SOTA baseline in ImageNet-100. We summarize our contributions as follows:

- We identify and formalize the optimization conflict in Non-Negative Contrastive Learning caused by common features, revealing that deterministic similarity measures are insufficient for perfect interpretability.

- We propose BayesNCL, a Bayesian Gating mechanism that allows the model to dynamically filter out

task-irrelevant common features while encouraging a principled sparse prior.

- We empirically show that BayesNCL significantly outperforms SOTA baselines in interpretability metrics (with a 142.1% gain in semantic consistency) without sacrificing representation power, offering a better alternative for reliable and interpretable self-supervised learning.

## 2. Related Work

**Contrastive learning.** Standard contrastive frameworks (Chen et al., 2020; He et al., 2020) excel at instance discrimination but implicitly assume images represent monolithic concepts (Wu et al., 2018; Saunshi et al., 2019). This assumption fails in real-world scenes composed of multiple semantic factors leading to the *compositionality gap* where semantically similar backgrounds are treated as negatives. To mitigate this, early approaches adopted *sample-level* strategies. For instance, Debiased Contrastive Learning (DCL) (Chuang et al., 2020) and Hard Negative Contrastive Learning (HCL) (Robinson et al., 2020) attempt to correct the training distribution by re-weighting false negatives or utilizing prototypes (Li et al., 2020). However, since these methods operate on entangled representations where semantic meanings are distributed across all dimensions, they must re-weight the entire image. To achieve granular control, recent work has shifted towards *feature-level disentanglement*. Most notably, Non-Negative Contrastive Learning (NCL) (Wang et al., 2024) enforces non-negativity to induce sparse representations where specific dimensions correspond to distinct semantic clusters. However, this explicit disentanglement hides a fundamental *optimization conflict*: when positive and negative pairs share a common feature, such as "blue sky", since NCL treats all activated dimensions with equal importance, the deterministic objective attempts to simultaneously align and repel the exact same feature dimension, causing gradient oscillation. This limitation motivates our proposed **BayesNCL**. By introducing a probabilistic gating mechanism with a Bayesian sparsity prior, BayesNCL explicitly models feature prevalence, adaptively filtering out uninformative commonalities during contrast to ensure semantic consistency. For more related works, please refer to Appendix B

## 3. Preliminaries

**Contrastive Learning.** We consider the standard representation learning setting where an encoder $f : \mathbb{R}^d \to \mathbb{R}^K$ extracts low-dimensional features $z = f(x) \in \mathbb{R}^K$ from inputs $x \in \mathbb{R}^d$. Data is generated via a three-step sampling process: first, a natural sample $\bar{x}$ is drawn from

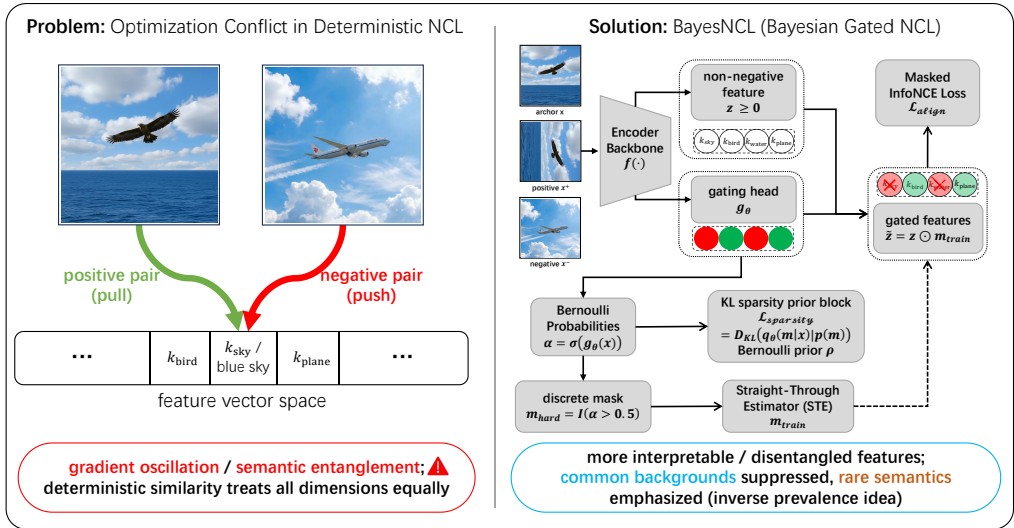

*Figure 2.* (Left) Deterministic similarity measure: When using deterministic similarity measures, common background features (e.g., "blue sky") shared between semantically distinct classes (e.g., "bird" and "plane") trigger opposing gradient objectives: alignment for positive pairs and repulsion for negative pairs. This conflict causes gradient oscillation and semantic entanglement. (Right) The BayesNCL Solution: To resolve this, we introduce a probabilistic gating mechanism. The gating head generates the Bernoulli parameter $\alpha$, which is thresholded to generate the discrete mask $m_{hard}$. We employ the Straight-Through Estimator to apply $m_{hard}$ during the forward pass ($z_{gated} = z \odot m_{hard}$) while allowing gradient backpropagation. The training objective combines a masked InfoNCE loss $L_{align}$ with a KL-divergence term $L_{sparsity}$, constraining the gate distribution against a sparse prior $\rho$ to suppress uninformative common features.

the data distribution $P(\bar{x})$; second, positive sample pair $(x, x^+)$ are sampled from the augmentation distribution $\mathcal{A}(\cdot|\bar{x})$; third, negative samples $x^-$ are drawn independently from the marginal distribution $P(x) = \mathbb{E}_{\bar{x}}[\mathcal{A}(x|\bar{x})]$. With $s(z, z') = z^\top z'/\tau$ denoting the temperature-scaled similarity between two representation vectors, the canonical InfoNCE loss (Oord et al., 2018) is formulated as:

$$\mathcal{L}_{\text{NCE}} = \mathbb{E}_{\mathcal{D}}\left[-s(z, z^+) + \log\left(e^{s(z,z^+)} + \sum_{j=1}^{M} e^{s(z,z_j^-)}\right)\right],$$

$$(1)$$

where $\mathcal{D}$ denote the data distribution sampling tuples of anchor $x$, positive $x^+$, and negatives $\{x_i^-\}_{i=1}^M$. Theoretically, this objective encourages the feature embeddings to be uniformly distributed on the hypersphere while maintaining local alignment for positive pairs (Wang & Isola, 2020).

**Non-negative Contrastive Learning (NCL).** Different from standard CL allows features to lie anywhere in the vector space, NCL (Wang et al., 2024) imposes a non-negative constraint on the encoder output, i.e., $f(x) \in \mathbb{R}_{\geq 0}^K$. By theoretically establishes the equivalence between the non-negative contrastive objective and Non-negative Matrix Factorization (NMF), NCL yields sparse and disentangled representations, in which each feature dimension tends to represent a distinct semantic attribute.

**Latent Class Formulation.** To analyze the semantic properties of the features, we strictly follow the latent class formulation introduced in NCL. Specifically, we assume the data is generated from a set of $m$ discrete latent classes $\mathcal{C} = \{c_1, \ldots, c_m\}$ with prior probabilities $P(c)$, and positive samples generated by contrastive learning are drawn independently from the same latent class. Ideally, if the feature dimension $k$ equals $m$, NCL proves that the optimal representation vector $\phi(x)$ approximates the posterior probability of latent classes up to a permutation and scaling:

$$\phi_j(x) \propto P(c_{\pi(j)}|x). \qquad (2)$$

This implies a strict correspondence where the $j$-th dimension of the feature vector indicates the membership of sample $x$ to a specific latent class.

## 4. Methodology

In this section, we first formalize the optimization conflict inherent in deterministic similarity measures. We then introduce **BayesNCL**, a variational framework that resolves this conflict via probabilistic gating. Finally, we provide theoretical analysis of our approach.

### 4.1. Problem Formulation: The Optimization Conflict

Standard Non-Negative Contrastive Learning (NCL) assumes a deterministic mapping $f : \mathcal{X} \to \mathbb{R}_{\geq 0}^K$. While

*Table 1.* Comparison of interpretability metrics between our approach and different baselines. ↑ indicates higher is better, ↓ indicates lower is better. "Act." represents the activation ratio of the feature dimension. **Bold** shows the superior result.

| Method | CIFAR-10 | | | | | CIFAR-100 | | | | | ImageNet-100 | | | | |
|---|---|---|---|---|---|---|---|---|---|---|---|---|---|---|---|
| | Cons. ↑ | $H_s$ ↓ | $H_m$ ↓ | $H_f$ ↓ | Act. | Cons. ↑ | $H_s$ ↓ | $H_m$ ↓ | $H_f$ ↓ | Act. | Cons. ↑ | $H_s$ ↓ | $H_m$ ↓ | $H_f$ ↓ | Act. |
| CL | 10.00 | 2.29 | 2.29 | 2.30 | 1.00 | 1.00 | 4.57 | 4.57 | 4.61 | 1.00 | 1.00 | 4.59 | 4.59 | 4.61 | 1.00 |
| NCL | 53.82 | 1.09 | **1.09** | 1.38 | 0.95 | 9.91 | 3.29 | 3.30 | 3.77 | 0.90 | 14.93 | 3.28 | 3.55 | 3.26 | 0.48 |
| NCL+TopK | 51.81 | 1.12 | 1.12 | 1.40 | 0.86 | 12.32 | 3.20 | 3.30 | 3.68 | 0.84 | 14.27 | 3.35 | 3.60 | 3.32 | 0.47 |
| BayesNCL (GS) | 53.68 | 1.12 | 1.16 | 1.28 | 0.63 | 20.75 | 2.81 | 3.33 | 3.20 | 0.84 | 11.66 | 3.36 | 3.40 | 3.70 | 0.13 |
| BayesNCL (STE) | **56.50** | **0.99** | 1.12 | **1.25** | 0.89 | **22.02** | **2.70** | **3.20** | **3.09** | 0.87 | **36.14** | **2.10** | **2.44** | **2.37** | 0.50 |

*Table 2.* Downstream Performance and Feature Efficiency. Comparison of Linear Probing Accuracy and Image Retrieval Precision. For Image Retrieval, we leverage the disentangled nature of our representations by performing feature selection, utilizing only the top $K$ dimensions ($K = 64$ for CIFAR-10; $K = 128$ for CIFAR-100; $K = 256$ for ImageNet-100). Since it is meaningless to select feature dimensions when using standard CL, we do not report the results. For all results, we report the mean of three random seeds.

| Method | CIFAR-10 | | | | CIFAR-100 | | | | ImageNet-100 | | | |
|---|---|---|---|---|---|---|---|---|---|---|---|---|
| | Linear Probe | | Retrieval | | Linear Probe | | Retrieval | | Linear Probe | | Retrieval | |
| | Acc@1 | Acc@5 | P@1 | P@3 | Acc@1 | Acc@5 | P@5 | P@10 | Acc@1 | Acc@5 | P@5 | P@10 |
| CL | 87.88 | 99.62 | - | - | 59.72 | 85.94 | - | - | 68.31 | 90.24 | - | - |
| NCL | 87.80 | 99.61 | 61.67 | 57.85 | 60.67 | 86.44 | 24.68 | 23.03 | 69.63 | 91.23 | 12.64 | 11.04 |
| NCL+TopK | **88.06** | **99.63** | 62.46 | 60.06 | **60.78** | 86.43 | 26.79 | 25.06 | 70.13 | 91.21 | 12.71 | 11.06 |
| BayesNCL | 88.02 | 99.59 | **63.59** | **61.23** | 60.69 | **86.62** | **28.47** | **26.61** | **70.44** | **91.71** | **13.13** | **11.32** |

effective for single-object images, this assumption fails in compositional scenes where objects share common background features, such as "blue sky". We define this phenomenon formally as the *Optimization Conflict*: the model simultaneously attempts to align shared features for positive pairs while repelling them for negative pairs. Our empirical analysis of NCL-pretrained representations reveals that feature dimensions with high entropy indeed frequently encode common background elements. These non-discriminative features induce spurious similarities between semantically distinct classes, providing evidence for the optimization conflict we aim to address (see Appendix J for more details).

Following the latent class formulation in NCL, a feature dimension $k$ is considered perfectly disentangled if it activates for exactly one class $c_i$. However, in real-world data, background features violate this exclusivity.

**Definition 4.1** (Conflicting Feature). A feature dimension $k$ is defined as a **Conflicting Feature** if it is active with high probability across a subset of semantically disjoint classes $S_k \subset \mathcal{C}$ where $|S_k| \geq 2$. Formally:

$$\exists c_i, c_j \in \mathcal{C}, i \neq j, \quad \text{s.t.} \quad P(z_k > \epsilon | c_i) \cdot P(z_k > \epsilon | c_j) \approx 1, \tag{3}$$

where $\epsilon$ is a small activation threshold.

To understand the consequence of conflicting features, we analyze the gradient of the standard InfoNCE loss with respect to a specific feature dimension $k$. The gradient can

be decomposed into two opposing forces:

$$\nabla_{z_k} \mathcal{L} = \underbrace{\frac{\partial \mathcal{L}_{pos}}{\partial s(z, z^+)} \cdot \frac{\partial s}{\partial z_k}}_{\mathcal{F}_{align,k}(<0)} + \sum_{j=1}^{M} \underbrace{\frac{\partial \mathcal{L}_{neg}}{\partial s(z, z_j^-)} \cdot \frac{\partial s}{\partial z_k}}_{\mathcal{F}_{rep,k}(>0)}, \tag{4}$$

where $s(\cdot, \cdot)$ denotes the similarity metric, such as the dot product commonly used in practice. $\mathcal{F}_{align,k}$ drives the feature to increase for positive pairs, while $\mathcal{F}_{rep,k}$ drives it to decrease for negative pairs. Such *conflicting features* (e.g., "blue sky") suffer from an optimization tug-of-war: the positive phase pulls the feature up via $\mathcal{F}_{align}$, whereas the negative phase pushes it down via $\mathcal{F}_{rep}$ due to the high prevalence of backgrounds in negative pairs. This conflict results in gradient oscillation and hinders semantic consistency.

**Proposition 4.2** (Gradient Instability). *For a conflicting feature $k$ defined in Definition 4.1, under the standard InfoNCE loss, the expected gradient approaches zero while the variance remains high:*

$$\mathbb{E}_{\mathcal{D}}[\nabla_{z_k} L] \approx 0, \quad \text{but} \quad Var_{\mathcal{D}}(\nabla_{z_k} L) \gg 0. \tag{5}$$

*Remark* 4.3. Proposition 4.2 indicates that deterministic similarity measures suffers from stochastic gradient oscillation on common dimensions. This prevents the optimizer from settling into a sparse solution, leading to entangled representations where background noise persists.

We summarize the corresponding feature-wise gradient evidence in Table 3, and provide the detailed empirical analysis in Section 5.

*Table 3.* **Gradient dynamics during pre-training on CIFAR-100.** We report Spearman correlations among Activation Frequency (AF), Gradient Variance (GV), and Semantic Consistency (SC) for each feature dimension.

| Epoch | NCL | | | BayesNCL (Ours) | | |
|---|---|---|---|---|---|---|
| | AF-GV | AF-SC | GV-SC | AF-GV | AF-SC | GV-SC |
| 10 | 0.748 | −0.880 | −0.646 | 0.337 | −0.954 | −0.268 |
| 20 | 0.753 | −0.861 | −0.675 | 0.285 | −0.919 | −0.189 |
| 30 | 0.787 | −0.906 | −0.688 | 0.178 | −0.893 | −0.070 |
| 40 | 0.728 | −0.869 | −0.627 | −0.066 | −0.887 | 0.177 |
| 50 | 0.816 | −0.906 | −0.706 | 0.326 | −0.866 | −0.189 |
| 60 | 0.705 | −0.912 | −0.596 | 0.469 | −0.836 | −0.339 |
| 70 | 0.694 | −0.881 | −0.607 | 0.519 | −0.858 | −0.398 |
| 80 | 0.703 | −0.882 | −0.605 | 0.644 | −0.869 | −0.468 |
| 90 | 0.635 | −0.880 | −0.522 | −0.008 | −0.789 | −0.001 |
| 100 | 0.735 | −0.910 | −0.643 | 0.563 | −0.863 | −0.399 |

**Motivation: Inverse-Prevalence Weighted Similarity.** To resolve this conflict, we revisit the definition of similarity from a probabilistic perspective and argue that the standard dot product is suboptimal for compositional data. Building on the latent class formulation, we derive the joint likelihood of a sample pair $(x, x')$ (see derivation in Appendix C) as $\log p(x, x') \propto \sum_k \frac{p(c_k|x)p(c_k|x')}{P(c_k)}$ . This motivates our proposed similarity metric:

**Definition 4.4** (Inverse-Prevalence Weighted Similarity). Let $\pi_k = P(c_k)$ denote the global prevalence of the $k$-th semantic factor. The **Inverse-Prevalence Weighted (IPW) Similarity** between two representations $z, z'$ is defined as:

$$s_{IPW}(z, z') = \sum_{k=1}^{K} \frac{1}{\pi_k} z_k z_k' = z^\top \mathbf{W} z', \qquad (6)$$

where $\mathbf{W} = \mathrm{diag}(1/\pi_1, \ldots, 1/\pi_K)$.

Definition 4.4 provides the theoretical justification for our method. It suggests that rare, discriminative features (low $\pi_k$) should dominate the alignment score, while high-frequency background features (high $\pi_k$) should be downweighted, which is consistent with the viewpoint of information theory (Shannon, 1948; Cover, 1999).

However, explicitly computing the scalar weight $1/\pi_k$ is intractable as semantic prevalence is unknown during training. BayesNCL achieves a tractable approximation to this theoretical optimum. By introducing a learnable gating mechanism, we dynamically suppress high-frequency features, effectively implementing Probability-Weighted Similarity via probabilistic filtering.

### 4.2. The BayesNCL Framework

To resolve the gradient instability, we propose **BayesNCL (Bayesian Gated Non-Negative Contrastive Learning)**. We transition from a deterministic view to a probabilistic one, where feature activation is modulated by a latent gating variable.

**Generative View.** Let $z = f(x) \in \mathbb{R}_{\geq 0}^K$ be the raw non-negative features. We introduce a binary latent mask $m \in \{0, 1\}^K$, which determines the subset of semantically relevant features. The joint distribution of a positive pair $(x, x^+)$ is defined by marginalizing over $m$:

$$p(x, x^+) \propto \int p(z, z^+|m)p(m)dm, \qquad (7)$$

where $p(m)$ is a sparsity-inducing prior, and $p(z, z^+|m)$ measures alignment solely on the gated subspace $\tilde{z} = z \odot m$.

**Variational Objective.** Since the marginalization is intractable, we employ amortized variational inference (Kingma & Welling, 2013; Rezende et al., 2014). We introduce a *Gating Head* parameterized by $\theta$ to approximate the posterior $q_\theta(m|x)$. Maximizing the Evidence Lower Bound (ELBO) yields our training objective (derivation in Appendix D):

$$\mathcal{L}_{\text{total}} = \underbrace{\mathbb{E}_{m \sim q_\theta}[\mathcal{L}_{\text{InfoNCE}}(z \odot m)]}_{\mathcal{L}_{\text{align}}} + \lambda \cdot \underbrace{D_{\text{KL}}(q_\theta(m|x)\|p(m))}_{\mathcal{L}_{\text{sparsity}}}.$$
$$(8)$$

**Implementation via Straight-Through Estimator.** The Gating Head predicts the Bernoulli parameters $\alpha = \sigma(g_\theta(x)) \in (0, 1)^K$, where $\sigma(\cdot)$ is sigmoid activation function. To enable backpropagation through the discrete sampling of $m$, we utilize the Straight-Through Estimator (STE) (Bengio et al., 2013). During forward pass, We compute a discrete mask to ensure strict feature selection: $m_{\text{hard}} = \mathbb{I}(\alpha > 0.5)$. During backward pass, gradients flow through the continuous probabilities: $m_{\text{train}} = \mathrm{sg}[m_{\text{hard}} - \alpha] + \alpha$, where $\mathrm{sg}[\cdot]$ denotes the stop-gradient operator. The prior $p(m)$ is set as a factorized Bernoulli distribution $\mathrm{Bern}(\rho)$, where $\rho$ is a hyperparameter controlling the expected sparsity.

### 4.3. Theoretical Analysis

In this section, we provide theoretical guarantees for BayesNCL from four complementary perspectives: local semantic filtering, global error reduction, information-theoretic optimality and generalization guarantees.

**Local Mechanism: Sparsity as Semantic Filtering.** The core innovation of BayesNCL is the $\mathcal{L}_{\text{sparsity}}$ term. We first prove that this term acts as an "Inverse Prevalence Filter", automatically suppressing features that appear too frequently, such as conflicting background features.

**Theorem 4.5** (Sparsity as Semantic Filtering). *Let $\pi_k$ denote the global prevalence of feature $k$ (i.e., the prior probability of it being active across the dataset). Minimizing the BayesNCL objective (Eq. 8) forces the optimal gating probability $\alpha_k^*$ to zero as $\pi_k \to 1$. Specifically, feature $k$ is gated out ($m_k = 0$) if:*

$$\underbrace{\gamma \cdot \pi_k(1 - \pi_k)}_{\text{Alignment Gain}} < \underbrace{\lambda \cdot \log(1/\rho)}_{\text{Sparsity Cost}}, \qquad (9)$$

*where $\gamma$ is a scaling factor related to the contrastive temperature.*

*Proof.* See Appendix E.1. □

*Remark* 4.6. Theorem 4.5 provides the local guarantee for solving the optimization conflict. Conflicting features (e.g., backgrounds) have high prevalence ($\pi_k \approx 1$), rendering their alignment gain negligible compared to the fixed sparsity cost. Consequently, BayesNCL automatically "switches off" these dimensions, halting the gradient oscillation described in Proposition 4.2.

**Global Guarantee: False Positive Error Reduction.** We now extend this local analysis to the entire feature space. Let the feature dimensions be partitioned into a Semantic Signal set $\mathcal{S}_{\text{signal}}$ (discriminative) and a Background Noise set $\mathcal{S}_{\text{noise}}$ (common, high prevalence). In standard contrastive learning, a major source of optimization conflict arises from *spurious alignment*: negative pairs that are semantically distinct but share common background features (e.g., a bird and a plane both in blue sky). We define the similarity score on these pairs as *Background-Induced False Positive Error*.

**Theorem 4.7** (Reduction of Background-Induced Error). *Consider a negative pair $(x, x^-)$ that is semantically disjoint but shares common background features in $\mathcal{S}_{noise}$. Let $E_{NCL}$ and $E_{Bayes}$ denote their expected similarity scores (error). Under the condition of Theorem 4.5, BayesNCL strictly reduces this error bound compared to deterministic NCL:*

$$\mathbb{E}[E_{Bayes}] \approx \mathbb{E}[E_{NCL}] - \underbrace{\sum_{k \in \mathcal{S}_{noise}} \mathbb{E}[z_k z_k^-]}_{\text{Spurious Similarity}}. \qquad (10)$$

*Proof.* See Appendix E.2. □

*Remark* 4.8. Theorem 4.7 leverages the fact that for semantically disjoint pairs, any similarity in $\mathcal{S}_{\text{noise}}$ constitutes a false positive. BayesNCL drives the gating probability of these high-prevalence features to zero, effectively pruning the spurious similarity term.

**Fundamental Principle: Information Constriction.** Finally, we analyze the information-theoretic implication of the gating mechanism. While standard Variational Information Bottleneck (VIB) (Alemi et al., 2016) methods minimize the mutual information $I(Z; X)$ by injecting stochastic noise, BayesNCL adopts a *structural* approach. By enforcing a sparse prior on the gating mask, we strictly limit the effective dimensionality of the representation space. This acts as a **Hard Information Bottleneck**, forcing the encoder to prioritize semantic content over high-frequency background noise.

**Theorem 4.9** (Information Bound via Sparsity). *Let $\tilde{Z} = Z \odot M$ be the gated representation, where $M \in \{0,1\}^K$ is the binary mask vector. Assuming the continuous features $Z$ are bounded within a finite volume, minimizing the sparsity regularizer $\mathcal{L}_{sparsity}$ imposes an upper bound on the mutual information $I(\tilde{Z}; X)$ via the expected activation rate:*

$$I(\tilde{Z}; X) \le \sum_{k=1}^{K} \mathbb{E}[m_k] \cdot C_{cont} + \mathcal{H}_b(\mathbb{E}[m_k]), \qquad (11)$$

*where $\mathbb{E}[m_k]$ is the activation probability of the $k$-th dimension, $C_{cont}$ is a constant bound on the differential entropy of active features, and $\mathcal{H}_b(\cdot)$ is the binary entropy function.*

*Proof.* See Appendix E.3. □

*Remark* 4.10. **Resolution of Optimization Conflict.** This theorem provides the theoretical justification for solving the "Bird/Plane" conflict. By restricting the channel capacity (sparsity), the model faces a resource allocation problem. To minimize the contrastive loss $\mathcal{L}_{\text{InfoNCE}}$, it *must* allocate the limited active dimensions to the most discriminative features (e.g., the bird's beak) while suppressing highly redundant features (e.g., the blue sky) that consume capacity without aiding instance discrimination.

## 5. Experiment

### 5.1. Experiment Setup

**Datasets.** We evaluate our methods on three benchmark datasets: cifar10, cifar100 (Krizhevsky et al., 2009) and Imagenet-100. Regarding ImageNet-100, since no official subset exists, we strictly adopt the partition established by Tian et al. (2020) to ensure fair and consistent comparisons.

**Baselines.** To evaluate the efficacy of our method, we compare it against standard Contrastive Learning (CL) and its non-negative variant mentioned in Section 3. Furthermore, to isolate the specific advantages of our probabilistic gating mechanism, we implement a Top-$k$ NCL baseline. This variant employs a deterministic hard-thresholding strategy, retaining only the $k$ highest-activating dimensions while masking others. For a fair comparison, the $k$ is chosen to remain consistent with the Bayesian prior $\rho$, i.e., $k = \text{round}(\rho \cdot D)$,

*Table 4.* Comparison of the effects of different gating mechanisms and gating head complexity. We report Interpretability metrics (Semantic Consistency, $H_f$) and Linear Probe performance (Acc@1, Acc@5). BayesNCL achieves the optimal trade-off. Full metrics are provided in Table 9 and Table 10 in Appendix H. For linear probe accuracy, we report the mean of three random seeds.

| | CIFAR-10 | | | | CIFAR-100 | | | | ImageNet-100 | | | |
|---|---|---|---|---|---|---|---|---|---|---|---|---|
| **Method** | *Interpretability* | | *Linear Probe* | | *Interpretability* | | *Linear Probe* | | *Interpretability* | | *Linear Probe* | |
| | Cons. ↑ | $H_f$ ↓ | Acc@1 | Acc@5 | Cons. ↑ | $H_f$ ↓ | Acc@1 | Acc@5 | Cons. ↑ | $H_f$ ↓ | Acc@1 | Acc@5 |
| *(a) Gating Mechanism* | | | | | | | | | | | | |
| Soft Gating | 51.84 | 1.40 | 87.72 | **99.62** | 8.54 | 3.86 | 60.71 | **86.83** | 33.15 | **2.26** | 69.87 | 91.47 |
| w/o Grad. Detach | 56.22 | 1.27 | **88.24** | 99.59 | 20.17 | 3.18 | 60.67 | 86.59 | 22.29 | 2.96 | 67.47 | 89.89 |
| *(b) Gating Head Complexity* | | | | | | | | | | | | |
| 1-Layer MLP | **56.97** | **1.22** | 87.87 | 99.57 | **26.70** | **2.75** | 60.34 | 86.02 | **40.85** | 2.38 | 69.55 | 90.63 |
| 3-Layer MLP | 55.26 | 1.29 | 87.87 | 99.50 | 17.88 | 3.33 | 60.60 | 86.34 | 26.37 | 2.83 | 68.09 | 90.43 |
| **BayesNCL (Ours)** | 56.50 | 1.25 | 88.02 | 99.59 | 22.02 | 3.09 | **60.73** | 86.62 | 36.14 | 2.37 | **70.44** | **91.71** |

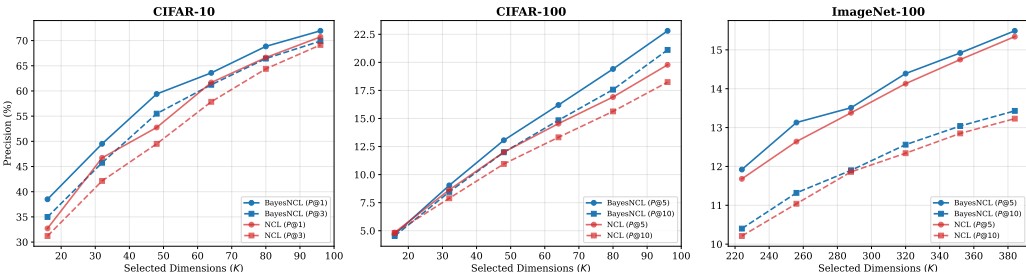

*Figure 3.* **Comparison of image retrieval performance on feature selection task.** The results show that compared to the NCL baseline (red), BayesNCL (blue) achieved consistent and significant performance improvement under different sparsity settings. This indicates that the features learned by BayesNCL have a higher semantic information density and can accurately represent image semantics through a very small number of key dimensions, validating its superior feature disentanglement capability. Detailed results are provided in Table 11 in Appendix H.

*Table 5.* **Scalability Verification on ResNet-50.** We report interpretability metrics to assess method performance on a deeper architecture.

| Method | Cons. ↑ | $H_s$ ↓ | $H_m$ ↓ | $H_f$ ↓ |
|---|---|---|---|---|
| CL | 1.00 | 4.59 | 4.59 | 4.61 |
| NCL | 6.39 | 3.82 | 3.82 | 4.07 |
| BayesNCL | **13.40** | **2.74** | **2.74** | **3.10** |

where $D$ denotes the total feature dimensionality. In addition, we also consider using Gumbel-Sigmoid (Jang et al., 2016; Maddison et al., 2016; Geng et al., 2020) as an alternative to STE in the gating mechanism.

**Evaluation Metrics.** To quantitatively assess the interpretability and disentanglement of the learned features, we employ two primary metrics. **Semantic Consistency (SC):** Measures whether an activated feature dimension is exclusively associated with a single semantic category. **Semantic Entropy (SE):** Captures the sparsity of the feature activation distribution across classes. A lower entropy indicates that a feature is specialized to a specific subset of classes. To ensure a comprehensive assessment, we report SE across

three distinct modalities: activation energy ($H_{\text{sum}}$), intensity ($H_{\text{mean}}$), and occurrence frequency ($H_{\text{freq}}$). Detailed mathematical definitions are provided in Appendix F.

**Implementation Details.** We adopt ResNet-18 (He et al., 2016) as the encoder backbone for all experiments. All models are optimized using the LARS optimizer (You et al., 2017). Due to space limitations, detailed hyperparameter settings are provided in **Appendix I**.

### 5.2. Main Results

We conduct a comprehensive evaluation of BayesNCL on multiple baselines. The results demonstrate the competitiveness of our approach and strongly support our hypothesis that probabilistic gating effectively resolves the optimization conflict in compositional data.

**Interpretability.** As reported in Table 1, BayesNCL establishes a new state-of-the-art in feature interpretability across all benchmarks. Notably, on ImageNet-100, our method achieves a Semantic Consistency score of **36.14**, representing a **142.1% relative improvement** over the standard NCL

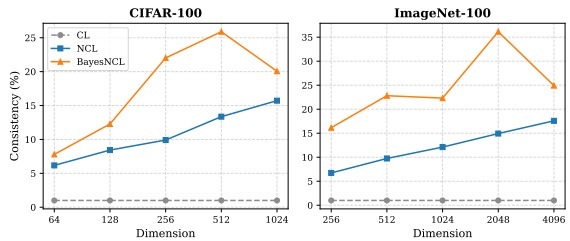
*(a)* Semantic Consistency with Feature Dimension Variation

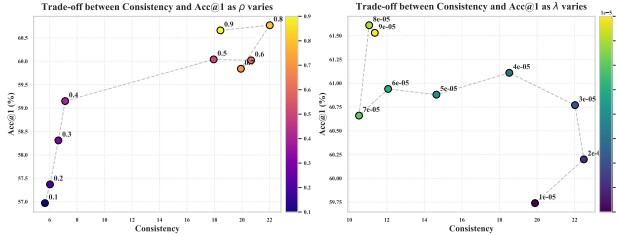
*(b)* Sensitivity to Sparsity Prior ($\rho$) and KL Weight ($\lambda$)

*Figure 4.* Analysis of Dimensional Scalability and Hyperparameter Trade-offs. **(a)** Consistency across varying feature dimensions. BayesNCL (orange) demonstrates superior scalability compared to NCL (blue) and CL (gray), effectively mitigating the optimization conflict in high-dimensional spaces on both CIFAR-100 and ImageNet-100. **(b)** The trade-off between Accuracy and Consistency under different configurations. The left plot varies the sparsity prior $\rho$ (Bernoulli parameter), while the right plot varies the KL regularization weight $\lambda$. The color gradient indicates the magnitude of the hyperparameter, highlighting the optimal operating region for balancing representation sparsity and discriminative power. Detailed results are provided in Table 12, Table 13 and Table 14 in Appendix H.

baseline (14.93%). This substantial gap confirms that the Bayesian gating mechanism successfully filters out polysemantic background features that typically entangle representations. To validate the gating discretization strategy, we compared our Straight-Through Estimator (STE) approach against the stochastic Gumbel-Sigmoid relaxation. We observe that Gumbel-Sigmoid underperforms STE, suggesting that deterministic gating is essential for the stability of the contrastive objective; stochastic exploration during the forward pass introduces variance that appears to confuse the model rather than aid disentanglement. Furthermore, we analyzed the Feature Activation Ratio to understand the source of these gains. Contrary to the intuition that interpretability stems solely from high sparsity, BayesNCL exhibits a higher activation ratio than NCL on ImageNet-100 while simultaneously achieving superior consistency. This indicates that BayesNCL does not achieve trivial sparsity by merely silencing channels. Instead, it promotes effective sparsity: it dynamically recruits more dimensions to encode distinct semantic concepts while successfully filtering out the entangled noise that plagues deterministic baselines. Moreover, referring to Table 5, the results on Resnet50 validate the scalability of our method.

**Representation Quality.** While interpretable models often suffer from a degradation in downstream task performance, BayesNCL remarkably breaks this trade-off. As shown in Table 2, our method maintains or exceeds the Linear Probe accuracy of the baselines. On ImageNet-100, BayesNCL achieves **70.44%** Top-1 accuracy, surpassing both SimCLR (68.31%) and NCL (69.63%). Besides, we also verified the quality of the representations using image retrieval on the feature selection task, and the results in Figure 3 show that our method outperforms NCL in multiple dimensions (for more details, please refer to Appendix H). We credit this performance gain to the "denoising" and "information concentration" capabilities of the gating mechanism. Specifically, by removing distracting high-frequency

features and compressing discriminative information into a smaller number of dimensions, the mechanism yields a cleaner representation vector with significantly enhanced representational power of feature vectors. In addition, we also visually analyzed the features suppressed by the gating head in Appendix K, and we found that the gating head indeed learned to suppress features that frequently appeared in the high-frequency background or puzzled the model.

**Background Preservation.** A natural concern is whether suppressing high-frequency common features damages the model's ability to represent background information. To test this, we evaluate frozen ImageNet-100-pretrained representations on background classification using Waterbirds (Sagawa et al., 2020), where a linear classifier is trained to predict the background label rather than the foreground object. As shown in Table 6, BayesNCL maintains a background classification accuracy comparable to NCL and even improves it slightly (93.38% vs. 93.33%). This indicates that BayesNCL does not discard coherent background information; instead, it preserves representational capacity while reducing the harmful foreground-background entanglement that causes optimization conflict.

*Table 6.* Linear probing accuracy for background classification on the Waterbirds dataset. The models are pre-trained on ImageNet-100.

| Method | Background Accuracy (%) |
|---|---|
| NCL | 93.33 |
| BayesNCL (Ours) | **93.38** |

**Gradient Dynamics.** To further test whether BayesNCL improves interpretability by resolving the optimization conflict proposed in Section 4, we analyze feature-wise gradient dynamics during pre-training on CIFAR-100. For each feature dimension, we compute Spearman correlations among Activation Frequency (AF), Gradient Variance (GV), and

Semantic Consistency (SC) every 10 epochs. As shown in Table 3, NCL exhibits a consistently strong positive AF-GV correlation, indicating that frequently activated dimensions tend to suffer larger gradient variance. At the same time, its negative GV-SC correlation shows that unstable gradients are associated with poorer semantic specialization. BayesNCL substantially weakens these harmful correlations, especially the AF-GV and GV-SC links, which supports our mechanism-level claim that probabilistic gating suppresses prevalent conflict-inducing features and stabilizes the learning of semantic dimensions.

**Computational Efficiency.** We also evaluate whether the Bayesian Gating Head and the Straight-Through Estimator introduce meaningful computational overhead. As shown in Table 7, BayesNCL adds only a small cost over NCL on both CIFAR-100 and ImageNet-100. For example, on CIFAR-100, the FLOPs increase only from 1.416G to 1.419G, while the training time increases from 70.95 to 75.12 minutes. Given the substantial gains in semantic consistency and the maintained downstream accuracy, this overhead represents a favorable trade-off between interpretability and efficiency.

*Table 7.* **Computational efficiency.** Comparison of training complexity and average running time between NCL and BayesNCL.

| Dataset | Method | Running Time (min) | FLOPs |
|---|---|---|---|
| CIFAR-100 | NCL | 70.95 | 1.416G |
| | BayesNCL (Ours) | 75.12 | 1.419G |
| ImageNet-100 | NCL | 193.78 | 3.731G |
| | BayesNCL (Ours) | 218.53 | 3.815G |

### 5.3. Ablation Study

In this section, we validate the design of BayesNCL, focusing on how probabilistic gating resolves the optimization conflict. Detailed experimental setups and additional analyses are provided in Appendix G.

**Gating Mechanism and Optimization Stability.** As shown in Table 4, we have thoroughly compared the different settings of gating mechanisms and the trade-offs between complexity of gating heads. We find that strict binary filtering is essential. Replacing the Hard Gating with Soft Gating causes a catastrophic drop in Semantic Consistency ($22.0\% \rightarrow 8.5\%$). This further confirms our hypothesis: merely down-weighting conflicting background features leaves residual gradients that sustain oscillation; strictly zeroing them out acts as a necessary circuit breaker. Furthermore, removing the gradient detachment degrades performance, indicating a *misalignment of objectives* where let the backbone minimize the sparsity loss $L_{sparsity}$ directly rather than learning robust representations. Finally, regarding the complexity of the gating head, we found that a 2-layer MLP is optimal capacity for the Gating Head, balancing non-linear selection capability with training stability.

**Dimensionality and Disentanglement Efficiency.** Inspired by SAE, which indicates the need for a sufficient number of feature channels for better disentanglement (Elhage et al., 2022; Cunningham et al., 2023; Li et al., 2026), we also investigate the relationship between latent width and semantic consistency in Figure 4a. We find that while NCL relies on expanding dimensions to mitigate feature superposition, it scales inefficiently. Instead, BayesNCL achieves superior disentanglement (36.14 consistency) at 2048 dimensions, twice that of NCL. This validates that dynamic probabilistic gating is a more parameter-efficient solution to the optimization conflict than simply increasing model capacity.

**Hyperparameter Sensitivity.** Sensitivity analysis on $\rho$ and $\lambda$ in Fig. 4b reveals an "Inverted-U" relationship. Moderate sparsity acts as a **semantic denoiser**, filtering common noise without discarding information. However, excessive regularization ($\lambda > 4e^{-5}$) forces a trade-off where the model sacrifices representation quality to satisfy the prior, highlighting the need for the KL term to act as a guide rather than a hard constraint. We also validate the learning rate of the Gating Head in Table 15, where the results show that setting it to $0.25\times$ the backbone learning rate is the most appropriate choice.

## 6. Conclusion

We identify that deterministic similarity measures in contrastive learning suffer from optimization conflicts, where shared background features induce semantic entanglement. To resolve this, we propose BayesNCL, which utilizes a variational gating mechanism to approximate *Inverse Prevalence Weighting*, dynamically suppressing high-frequency noise. Our approach achieves state-of-the-art semantic consistency on ImageNet-100, demonstrating that probabilistic constraints can resolve the performance-interpretability trade-off, paving the way for rigorous and disentangled Self-Supervised Learning.

## Acknowledgements

We thank the anonymous reviewers for their valuable comments and suggestions. This work is partially supported by the MBZUAI Research Fund BF0100.

## Impact Statement

This paper presents work whose goal is to advance the field of Machine Learning. There are many potential societal consequences of our work, none which we feel must be specifically highlighted here.

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

# A. Notation

Table 8 summarizes the mathematical notations used throughout this paper. We categorize these symbols into three groups: (1) **General Framework**, which covers the standard contrastive learning setup and latent class assumptions; (2) **Bayesian Gating Mechanism**, which introduces the variables specific to BayesNCL used to resolve the *Optimization Conflict* via probabilistic filtering; and (3) **Optimization & Metrics**, detailing the loss components and hyperparameters governing the sparsity prior.

*Table 8.* Summary of Notations.

| Symbol | Description |
|---|---|
| *General Framework & Latent Semantics* | |
| $x, \bar{x}$ | An input image and its underlying natural source (before augmentation). |
| $x^+, x^-$ | Positive sample (augmented view of $x$) and negative sample (independent instance). |
| $f(\cdot)$ | The encoder backbone. |
| $z \in \mathbb{R}_{\geq 0}^K$ | The raw, non-negative feature representation. |
| $C = \{c_1, ..., c_K\}$ | The set of discrete latent semantic classes (e.g., objects, attributes). |
| $K$ | The dimensionality of the feature space. |
| $\phi(x)$ | The theoretical optimal representation vector approximating the posterior $P(c|x)$. |
| $\pi_k$ | **Prevalence** of the $k$-th latent class, defined as global prior $P(c_k)$. High $\pi_k$ indicates common features (e.g., background noise) causing optimization conflicts. |
| *Bayesian Gating Mechanism (Ours)* | |
| $g_\theta(\cdot)$ | The **Gating Head**, a lightweight MLP parameterized by $\theta$. |
| $\alpha \in (0,1)^K$ | The predicted Bernoulli parameters (soft gating probabilities) for each dimension, where $\alpha = \sigma(g_\theta(x))$. |
| $m \in \{0,1\}^K$ | The binary gating mask sampled from the posterior distribution. |
| $m_{hard}$ | The discretized mask used for the forward pass, calculated as $\mathbb{I}(\alpha > 0.5)$. |
| $m_{train}$ | The mask used during backpropagation via the **Straight-Through Estimator (STE)**, allowing gradients to flow through discrete decisions. |
| $\tilde{z}$ | The **Gated Feature Representation**, computed as $\tilde{z} = z \odot m_{train}$. This is the input to the contrastive loss, free from conflicting high-frequency noise. |
| *Optimization & Metrics* | |
| $\mathcal{L}_{align}$ | The Masked InfoNCE loss, calculated using gated features $\tilde{z}$. |
| $\mathcal{L}_{sparsity}$ | The KL-divergence regularization term forcing the mask distribution towards the prior. |
| $\rho$ | The **Sparsity Prior**, a hyperparameter defining the expected activation rate (Bernoulli parameter) of the prior distribution $P(m)$. |
| $\lambda$ | Weighting coefficient for the sparsity regularization term. |
| $\tau$ | Temperature parameter for the contrastive loss. |
| $SC_j, SE_j$ | Semantic Consistency and Semantic Entropy metrics for the $j$-th feature dimension. |

# B. Related Works

While the main text focuses on the intersection of non-negativity and optimization conflicts, the paradigm of Contrastive Learning (CL) has evolved through several distinct phases. In this section, we provide a broader historical perspective, tracing the trajectory from statistical estimation principles to modern dense and multimodal frameworks.

**Theoretical Origins: From NCE to InfoNCE**   The mathematical foundation of modern contrastive learning predates the deep learning boom. It originates from **Noise Contrastive Estimation (NCE)** (Gutmann & Hyvärinen, 2010), proposed as a method to estimate energy-based models by discriminating observed data from noise samples, avoiding the calculation of intractable partition functions. This principle was first popularized in representation learning by **Word2Vec** (Mikolov et al., 2013), which utilized negative sampling to efficiently learn semantic word vectors. In the visual domain, early self-supervised approaches such as **Context Encoders** (Pathak et al., 2016) and **Colorization** (Zhang et al., 2016) implicitly relied on contrastive signals to solve pretext tasks. However, the pivotal moment for modern CL was the introduction of **Contrastive Predictive Coding (CPC)** (Oord et al., 2018), which generalized NCE into the widely adopted **InfoNCE** objective, establishing a universal framework for mutual information maximization.

**The Instance Discrimination Era** The field subsequently shifted from specific pretext tasks to the generalized paradigm of **Instance Discrimination** (Wu et al., 2018), where every image is treated as its own distinct class. A primary challenge in this era was scaling negative samples efficiently. **MoCo** (He et al., 2020) addressed this via a momentum-updated queue, decoupling the negative sample size from the mini-batch size. Conversely, **SimCLR** (Chen et al., 2020) demonstrated that end-to-end training was feasible given sufficiently large batches and strong augmentations, such as Gaussian blur and color jitter. Following these architectural advances, research began to focus on the quality of the contrastive signal. Approaches like **Hard Negative Mixing** (Kalantidis et al., 2020) and HCL (Robinson et al., 2020) identified that not all negatives are equal, proposing strategies to synthesize or up-weight harder negatives. While these methods improved gradient signaling, they largely operated at the sample level, a limitation our BayesNCL addresses by identifying "conflicting" features within the representation itself.

**Beyond Global Features: Dense and Pixel-Level CL** Standard contrastive frameworks typically pool the entire image into a single global vector, which inevitably destroys spatial information and fine-grained details. To adapt contrastive learning for dense prediction tasks like segmentation and detection, the field evolved towards local contrast strategies. **DenseCL** (Wang et al., 2021) and **PixPro** (Xie et al., 2021b) extended the paradigm to the pixel or patch level, ensuring that local features remain consistent across views. Similarly, **DetCo** (Xie et al., 2021a) introduced hierarchical contrastive learning specifically optimized for object detection. Crucially, these dense methods face an exacerbated version of the "Optimization Conflict" we identify in the main text: local patches often consist entirely of background elements (e.g., sky, grass), making the disentanglement of common features from semantic objects even more critical to prevent false repulsion.

**Multimodal Contrastive Learning** Perhaps the most impactful expansion of CL has been the alignment of vision with language. Models such as **CLIP** (Radford et al., 2021) and **ALIGN** (Jia et al., 2021) scaled the contrastive objective to massive web-scale datasets, learning to align image embeddings with corresponding text embeddings, while **SLIP** (Mu et al., 2022) further combined self-supervised image contrast with image-text supervision. In these multimodal settings, the polysemantic nature of visual features becomes a major bottleneck, for instance, a caption may mention a dog, but the image contains both a dog and grass. This creates a misalignment where the visual embedding contains information irrelevant to the text. Our BayesNCL gating mechanism offers a theoretical pathway to cleaner cross-modal alignment by potentially filtering non-corresponding visual tokens during the alignment process.

## C. Deriving the Inverse-Prevalence Weighted Similarity

To formally understand how overlapping features affect representation learning, we revisit the joint probability of a sample pair $(x, x')$ based on the latent class model. By decomposing the joint distribution via latent classes $\mathcal{C}$, we have:

$$\begin{aligned}
p(x, x') &= \sum_{c \in \mathcal{C}} p(x, x'|c)p(c) = \sum_{c \in \mathcal{C}} p(x|c)p(x'|c)p(c) \\
&= \sum_{c \in \mathcal{C}} \frac{p(c|x)p(x)}{p(c)} \frac{p(c|x')p(x')}{p(c)} p(c) \\
&= p(x)p(x') \sum_{c \in \mathcal{C}} \frac{p(c|x)p(c|x')}{P(c)}.
\end{aligned} \tag{12}$$

Taking the logarithm, the log-likelihood of observing the pair $(x, x')$ decomposes into marginal priors and an interaction term:

$$\log p(x, x') = \underbrace{\log p(x) + \log p(x')}_{\text{Sample Priors}} + \underbrace{\log \left( \sum_{k=1}^{m} \frac{p(c_k|x)p(c_k|x')}{P(c_k)} \right)}_{\text{Semantic Alignment}}. \tag{13}$$

In the context of NCL, the features are interpreted as posterior probabilities, i.e., $\phi_k(x) \approx p(c_k|x)$. Let $\pi_k = P(c_k)$ denote the global prior probability (prevalence) of the $k$-th latent class. The interaction term suggests a theoretical form for **inverse-prevalence weighted similarity**:

$$\mathcal{S}_{\text{ideal}}(x, x') = \sum_{k=1}^{m} \frac{\phi_k(x)\phi_k(x')}{\pi_k}. \tag{14}$$

Eq. 14 reveals a fundamental principle from Information Theory: rarer features should carry higher weight.

- For **distinctive semantics** (e.g., specific objects), $\pi_k$ is small, amplifying their contribution to similarity.

- For **common features** (e.g., generic 'blue sky'), $\pi_k$ is large, necessitating a down-weighting mechanism to reduce their dominance.

## D. Derivation of the Variational Objective

In this section, we provide the rigorous derivation of the BayesNCL objective function from the perspective of variational inference.

### D.1. Generative Model Setup

We assume the data generation process for a positive pair $(x, x^+)$ is governed by a latent binary mask variable $m \in \{0, 1\}^K$. The generative process is defined as follows:

1. **Prior:** The latent mask $m$ is drawn from a sparse Bernoulli prior:

$$m \sim p(m) = \prod_{k=1}^{K} \text{Bern}(m_k; \rho), \tag{15}$$

where $\rho$ is the sparsity hyperparameter.

2. **Likelihood:** Given the mask $m$, the feature representations $z = f(x)$ and $z^+ = f(x^+)$ are generated such that their similarity is maximized on the active dimensions. We model the conditional likelihood of the positive sample $z^+$ given the anchor $z$ and mask $m$ using a log-linear model (related to the Von Mises-Fisher distribution on the hypersphere):

$$p(z^+|z, m) \propto \exp\left(\frac{(z \odot m)^\top (z^+ \odot m)}{\tau}\right), \tag{16}$$

where $\odot$ denotes the element-wise product and $\tau$ is the temperature parameter.

### D.2. Evidence Lower Bound (ELBO)

Our goal is to maximize the marginal log-likelihood of observing the positive pairs, marginalizing over the latent mask $m$. Since the integral $\log \int p(z^+|z, m)p(m)dm$ is intractable, we introduce a variational posterior $q_\theta(m|z)$ (parameterized by the Gating Head) to approximate the true posterior.

Using Jensen's Inequality, we derive the Evidence Lower Bound (ELBO):

$$\log p(z^+|z) = \log \int p(z^+|z, m)p(m)dm \tag{17}$$

$$= \log \int q_\theta(m|z) \frac{p(z^+|z, m)p(m)}{q_\theta(m|z)} dm \tag{18}$$

$$\geq \int q_\theta(m|z) \log\left(\frac{p(z^+|z, m)p(m)}{q_\theta(m|z)}\right) dm \quad \text{(Jensen's Inequality)} \tag{19}$$

$$= \mathbb{E}_{m \sim q_\theta}[\log p(z^+|z, m)] - D_{KL}(q_\theta(m|z)\|p(m)) \tag{20}$$

$$= -\mathcal{L}_{reconstruction} - \mathcal{L}_{regularization}. \tag{21}$$

### D.3. Connecting to BayesNCL Loss

We now map the terms in the ELBO to our specific loss functions.

**1. The Sparsity Term ($\mathcal{L}_{sparsity}$):** The second term is explicitly the KL divergence between the predicted mask distribution and the Bernoulli prior. Since we assume factorized distributions, this matches Eq. 10 in the main text:

$$\mathcal{L}_{regularization} = \sum_{k=1}^{K} D_{KL}(\text{Bern}(\alpha_k) \| \text{Bern}(\rho)). \tag{22}$$

**2. The Alignment Term ($\mathcal{L}_{align}$):** The first term maximizes the expected log-likelihood of the positive pair under the mask. In Contrastive Learning, directly maximizing the likelihood is difficult due to the partition function. However, it is well-established that the InfoNCE loss is a lower bound on the mutual information, which effectively maximizes this log-likelihood ratio against noise samples:

$$\mathbb{E}_{m \sim q_\theta}[\log p(z^+|z, m)] \approx -\mathcal{L}_{InfoNCE}(\tilde{z}, \tilde{z}^+), \tag{23}$$

where $\tilde{z} = z \odot m$. By utilizing the Straight-Through Estimator (STE), we approximate the expectation over $m \sim q_\theta$ by using the hard mask $m_{hard}$ during the forward pass while backpropagating gradients through the soft probabilities $\alpha$.

Combining these, minimizing the negative ELBO is equivalent to minimizing our total loss:

$$\mathcal{L}_{total} = \mathcal{L}_{align} + \lambda \mathcal{L}_{sparsity}. \tag{24}$$

This concludes the derivation.

# E. Theoretical Proof

### E.1. Proof of Theorem 4.5

*Proof.* Consider a single feature dimension $k$ with global prevalence $\pi_k = P(m_k = 1)$. Let the local objective function for this dimension be $J_k = \Delta L_{align}(k) - \lambda \Delta L_{sparsity}(k)$, where $\Delta$ denotes the contribution of dimension $k$ to the total loss.

**Utility Analysis ($\Delta L_{align}$):** In contrastive learning, a feature $k$ is discriminative if it is present in the positive pair $(x, x^+)$ but absent in the negative samples $\{x_j^-\}$. Let $U_k$ be the utility of feature $k$. The probability that feature $k$ successfully discriminates a positive pair from a random negative sample is:

$$P(\text{Discrimination}_k) \propto P(k \in x, k \in x^+) \cdot P(k \notin x^-) \approx \pi_k \cdot (1 - \pi_k). \tag{25}$$

Thus, the expected information gain $\Delta L_{align}(k)$ is proportional to $\pi_k(1 - \pi_k)$. Note that as $\pi_k \to 1$ (common backgrounds) or $\pi_k \to 0$ (extremely rare noise), the utility $U_k$ approaches zero.

**Cost Analysis ($\Delta L_{sparsity}$):** The sparsity term penalizes the divergence between the posterior activation $\alpha_k$ and the sparse prior $\rho$. For a feature that the model potentially keeps active ($\alpha_k \approx 1$), the cost is:

$$\Delta L_{sparsity}(k) = D_{KL}(\text{Bern}(1) \| \text{Bern}(\rho)) = \log \frac{1}{\rho}. \tag{26}$$

Since $\rho$ is a constant less than 1, this cost is always a positive penalty.

**Optimal Gating Decision:** The gating head will activate dimension $k$ (i.e., $m_k = 1$) only if the utility outweighs the cost:

$$\gamma \cdot \pi_k(1 - \pi_k) > \lambda \cdot \log \frac{1}{\rho}, \tag{27}$$

where $\gamma$ is a scaling factor. For high-prevalence background features where $\pi_k \to 1$, the term $(1 - \pi_k)$ vanishes, leading to:

$$\lim_{\pi_k \to 1} \gamma \cdot \pi_k(1 - \pi_k) = 0 < \lambda \cdot \log \frac{1}{\rho}. \tag{28}$$

In this regime, the inequality fails, and the optimal variational posterior $q_\theta(m_k|x)$ collapses to 0. This proves that BayesNCL mathematically necessitates the suppression of high-frequency common features, effectively approximating the Inverse Prevalence Weighting $1/\pi_k$ via hard thresholding. $\square$

### E.2. Proof of Theorem 4.7

**Problem Setup & Definitions.** Let $(x, x^-)$ be a negative pair sampled during training. To rigorously define "False Positive Error," we assume the existence of latent semantic labels $y, y^-$. We focus on the specific case of **Background-Induced Hard Negatives**, defined by two conditions:

1. **Semantically Disjoint:** $y \neq y^-$ (e.g., Bird vs. Plane). Ideally, their similarity should be zero.

2. **Background Overlap:** They share high activation on nuisance features $\mathcal{S}_{\text{noise}}$ (e.g., Blue Sky).

Based on Theorem 4.5, we partition the feature space into Semantic Signal $\mathcal{S}_{\text{signal}}$ (low prevalence) and Background Noise $\mathcal{S}_{\text{noise}}$ (high prevalence).

**Error in Deterministic NCL.** In standard NCL, features are non-negative ($z \geq 0$). The similarity score (Error) for this disjoint pair is:

$$E_{\text{NCL}}(x, x^-) = \underbrace{\sum_{k \in \mathcal{S}_{\text{signal}}} z_k z_k^-}_{\text{Semantic Coincidence}} + \underbrace{\sum_{k \in \mathcal{S}_{\text{noise}}} z_k z_k^-}_{\text{Background Spuriousness}} . \tag{29}$$

Since $y \neq y^-$, the semantic coincidence term is expected to be small (assuming disentanglement). However, since $k \in \mathcal{S}_{\text{noise}}$ represents high-prevalence common features, the term $\sum_{k \in \mathcal{S}_{\text{noise}}} z_k z_k^-$ is strictly positive and large. This high similarity constitutes a significant *False Positive Error*, causing the model to pull apart the "Blue Sky" feature, which conflicts with positive pairs that share the same sky.

**Error in BayesNCL.** BayesNCL computes similarity on $\tilde{z} = z \odot m$. The expected error is:

$$\mathbb{E}[E_{\text{Bayes}}] = \sum_{k \in \mathcal{S}_{\text{signal}}} \mathbb{E}[\alpha_k \alpha_k^-] z_k z_k^- + \sum_{k \in \mathcal{S}_{\text{noise}}} \mathbb{E}[\alpha_k \alpha_k^-] z_k z_k^- . \tag{30}$$

**Applying Theorem 4.5:** For features $k \in \mathcal{S}_{\text{noise}}$, the global prevalence $\pi_k$ is high. Theorem 4.5 proves that the sparsity constraint forces the gating probability $\alpha_k(x)$ to approach zero for such features. Let $\alpha_k(x) \leq \epsilon$ for $k \in \mathcal{S}_{\text{noise}}$. The noise term becomes:

$$\sum_{k \in \mathcal{S}_{\text{noise}}} \mathbb{E}[\alpha_k \alpha_k^-] z_k z_k^- \leq \epsilon^2 \sum_{k \in \mathcal{S}_{\text{noise}}} z_k z_k^- \approx 0. \tag{31}$$

**Conclusion.** Comparing the two errors:

$$\Delta E = \mathbb{E}[E_{\text{NCL}}] - \mathbb{E}[E_{\text{Bayes}}] \approx \sum_{k \in \mathcal{S}_{\text{noise}}} z_k z_k^- . \tag{32}$$

Since $z \geq 0$, this difference is strictly positive. This proves that BayesNCL specifically suppresses the similarity contribution arising from high-prevalence background features. For semantically disjoint pairs ($y \neq y^-$), this directly translates to a reduction in False Positive Error, thereby mitigating the optimization conflict. $\qquad\square$

### E.3. Proof of Theorem 4.9

**Goal.** We demonstrate that minimizing $\mathcal{L}_{\text{sparsity}}$ minimizes an upper bound on the mutual information $I(\tilde{Z}; X)$.

**Decomposition of Mutual Information.** Since $\tilde{Z}$ is a deterministic function of $Z$ and $M$, and $M$ is conditioned on $X$, we analyze the joint information. The mutual information is upper bounded by the joint entropy of the representation and the mask:

$$I(\tilde{Z}; X) \leq H(\tilde{Z}) \leq H(\tilde{Z}, M). \tag{33}$$

Using the chain rule of entropy, we decompose $H(\tilde{Z}, M)$:

$$H(\tilde{Z}, M) = H(M) + H(\tilde{Z}|M). \tag{34}$$

**Bounding the Discrete Entropy** $H(M)$**.** Assuming dimensions are conditionally independent given $X$ (mean-field approximation used in our variational posterior), the entropy of the mask is the sum of binary entropies. Let $p_k = P(m_k = 1|X)$.

$$H(M) \leq \sum_{k=1}^{K} \mathcal{H}_b(p_k), \tag{35}$$

where $\mathcal{H}_b(p) = -p \log p - (1-p) \log(1-p)$. As $p_k \to 0$ (enforced by sparsity), $\mathcal{H}_b(p_k) \to 0$.

**Bounding the Conditional Differential Entropy** $H(\tilde{Z}|M)$**.** The term $H(\tilde{Z}|M)$ represents the entropy of the features *given* their activation status.

$$H(\tilde{Z}|M) = \sum_{k=1}^{K} \left[ P(m_k = 0)H(\tilde{Z}_k|m_k = 0) + P(m_k = 1)H(\tilde{Z}_k|m_k = 1) \right]. \tag{36}$$

- If $m_k = 0$, then $\tilde{z}_k = 0$ deterministically. The entropy is 0 (or $-\infty$ in differential terms, but effectively 0 information content relative to the subspace).

- If $m_k = 1$, then $\tilde{z}_k = z_k$. Since we employ normalization and bounded activation functions (ReLU/Sigmoid) in the projector, the differential entropy of active features is bounded by a constant $C_{\text{cont}}$ (related to the volume of the feature support).

Thus:

$$H(\tilde{Z}|M) \leq \sum_{k=1}^{K} p_k \cdot C_{\text{cont}}. \tag{37}$$

**The Role of** $\mathcal{L}_{\textbf{sparsity}}$**.** Our regularization term minimizes $D_{\text{KL}}(\text{Bern}(p_k)\|\text{Bern}(\rho))$. This forces the posterior activation probability $p_k$ towards the small prior $\rho$. Combining the terms:

$$I(\tilde{Z}; X) \leq \sum_{k=1}^{K} \underbrace{\mathcal{H}_b(p_k)}_{\text{Mask Cost}} + \underbrace{p_k \cdot C_{\text{cont}}}_{\text{Feature Cost}}. \tag{38}$$

Both terms are monotonically increasing with $p_k$ (for $p_k < 0.5$). Therefore, minimizing $\mathcal{L}_{\text{sparsity}}$ directly minimizes the upper bound on mutual information.

**Conclusion.** BayesNCL implements a bottleneck by restricting the **expected number of active dimensions** ($\sum p_k$). Unlike standard NCL which allows information to flow through all $K$ channels, BayesNCL constricts the flow to $\approx K\rho$ channels. This mathematically necessitates the "Inverse Prevalence Weighting" effect: to maintain low contrastive loss under this tight capacity constraint, the encoder must discard high-prevalence, low-information features (backgrounds) in favor of rare, high-information features (objects). □

# F. Detailed Evaluation Metrics

To quantitatively evaluate the alignment between learned latent dimensions and semantic categories, we employ Semantic Consistency (SC) and Semantic Entropy (SE). Let $\mathcal{D} = \{(x_i, y_i)\}_{i=1}^{N}$ denote the dataset with $C$ classes. We define a feature dimension $j$ as *active* for a sample $x$ if its absolute activation magnitude exceeds a threshold $\epsilon = 10^{-5}$, denoted by the indicator $\mathbb{I}(|z_j(x)| > \epsilon)$. Let $\mathcal{S}_j = \{(x,y) \in \mathcal{D} \mid |z_j(x)| > \epsilon\}$ be the set of samples where dimension $j$ is active.

### F.1. Semantic Consistency (SC)

Following Wang et al. (2024), SC measures the dominance of the most frequent class for a given feature dimension $j$:

$$\text{SC}_j = \max_{c \in \{1,\dots,C\}} \frac{\sum_{(x,y) \in \mathcal{S}_j} \mathbb{I}(y = c)}{|\mathcal{S}_j|}$$

Intuitively, SC determines if a feature is a "class detector."

### F.2. Semantic Entropy (SE)

While SC focuses only on the dominant class, Semantic Entropy captures the full distribution of feature activations. We define SE as the entropy of the class distribution $p_j$ for dimension $j$:

$$\text{SE}_j(p) = -\sum_{c=1}^{C} p_{j,c} \log p_{j,c}$$

To analyze feature semantics from different physical perspectives (energy, intensity, and frequency), we instantiate the distribution $p_j$ in three ways:

- **SE-Sum ($H_{\text{sum}}$):** Measures the concentration of total activation energy.

$$p_{j,c}^{(\text{sum})} = \frac{\sum_{(x,y)\in\mathcal{S}_j, y=c} |z_j(x)|}{\sum_{(x',y')\in\mathcal{S}_j} |z_j(x')|}$$

- **SE-Mean ($H_{\text{mean}}$):** Isolates average activation intensity, normalizing for class imbalance.

$$p_{j,c}^{(\text{mean})} = \frac{\bar{z}_{j,c}}{\sum_{k=1}^{C} \bar{z}_{j,k}}$$

  where $\bar{z}_{j,c} = \frac{1}{N_c} \sum_{(x,y)\in\mathcal{S}_j, y=c} |z_j(x)|$ and $N_c$ is the number of samples in class $c$.

- **SE-Freq ($H_{\text{freq}}$):** Assesses the breadth of feature occurrence (prevalence). This is particularly sensitive to "common features" (e.g., background textures) that create optimization conflicts.

$$p_{j,c}^{(\text{freq})} = \frac{\sum_{(x,y)\in\mathcal{S}_j} \mathbb{I}(y=c)}{|\mathcal{S}_j|}$$

For all metrics, we report the mean value across all active dimensions $\mathcal{A}$, i.e., $\text{SE} = \frac{1}{|\mathcal{A}|} \sum_{j\in\mathcal{A}} \text{SE}_j$.

## G. Detailed Analysis of Ablation Experiments

We conduct extensive ablations to validate the design choices of the Bayesian Gating mechanism, specifically investigating how discreteness, gradient flow, and model capacity influence the resolution of the optimization conflict.

**Is strict binary filtering necessary?** (Table 4 & 9). We investigate whether the gating mechanism requires a hard threshold or if soft down-weighting is sufficient. We compare our default Hard Gating (via Straight-Through Estimator) against a Soft Gating variant using continuous sigmoid activations. Results indicate that **Hard Gating is non-negotiable** for interpretability. The Soft Gating variant suffers a catastrophic drop in Semantic Consistency (8.54% vs. 22.02% on CIFAR-100). This validates our core hypothesis regarding the optimization conflict: merely down-weighting conflicting features (e.g., background noise) leaves residual gradients that sustain oscillation. Strictly zeroing out these dimensions ($z_{gated} = z \odot m_{hard}$) acts as a necessary circuit breaker, forcing the model to rely solely on discriminative semantics.

**Is gradient detachment essential for stable optimization?** (Table 4 & 9). We further analyze the necessity of detaching gradients before the Gating Head input. In the "w.o. detach" variant, we allow gradients from the KL sparsity loss to backpropagate into the backbone encoder. The results show that removing the detach operation degrades both consistency and accuracy. We attribute this to a *misalignment of objectives*: without detachment, the backbone learns to suppress feature activations globally to trivially satisfy the sparsity prior $L_{sparsity}$, rather than allowing the Gating Head to learn intelligent selection policies. The detach operation ensures the backbone focuses solely on representation learning, while the Gating Head specializes in semantic filtering.

**What is the optimal complexity for the Gating Head?** (Tables 4 & 10). We explore whether the gating policy requires deep non-linear modeling. Empirically, a **2-layer MLP** strikes the optimal balance. While a single linear layer yields high consistency, it causes a notable drop in representation quality (Acc@1), suggesting that linear gating is overly aggressive and may prune informative but non-linear feature dependencies. Conversely, a 3-layer MLP degrades performance on

both metrics, likely due to optimization difficulties inherent in training deeper networks with discrete bottlenecks. Thus, a lightweight 2-layer architecture provides sufficient capacity to model feature conditional probabilities without inducing training instability.

**Does increasing feature dimensionality enhance semantic disentanglement?** (Table 12). Recent findings in Sparse Autoencoders (SAEs) suggest that semantic concepts in neural networks are often polysemantic due to the superposition of features in limited dimensions (Elhage et al., 2022; Cunningham et al., 2023; Li et al., 2026). Motivated by this, which suggests wider layers alleviate polysemanticity, we examine consistency across varying dimensions. Standard CL remains rotationally invariant with flat consistency scores, confirming its inability to align dimensions with semantics. NCL shows monotonic improvement as width increases, e.g., $6.73 \rightarrow 17.58$ on ImageNet-100, validating that enforcing non-negativity utilizes capacity to disentangle concepts. However, BayesNCL demonstrates superior efficiency, achieving a peak consistency of 36.14 at 2048 dimensions on ImageNet-100—more than double that of NCL. This confirms that probabilistic gating resolves feature conflicts more effectively than simply expanding the latent space, allowing higher disentanglement with fewer parameters.

**How do sparsity constraints modulate the trade-off?** (Table 13 & 14). Finally, we analyze the sensitivity to the sparsity prior $\rho$ and KL weight $\lambda$. Figure 4b reveals an "Inverted-U" relationship for the sparsity prior $\rho$. Performance peaks at moderate sparsity; extreme sparsity ($\rho < 0.4$) destroys essential information, while high density ($\rho \rightarrow 0.9$) re-introduces entangled background noise. Similarly, regarding the KL weight $\lambda$, moderate penalties act as a **semantic denoiser**, improving both accuracy and consistency. However, excessive regularization ($\lambda > 4e - 5$) creates a new conflict where the model sacrifices interpretability to maintain representation quality. This confirms that BayesNCL operates best when the prior serves as a gentle guide rather than a hard constraint.

## H. Additional Experimental Results

*Table 9.* Comparing the Impact of different gating mechanisms. $\uparrow$ indicates higher is better, $\downarrow$ indicates lower is better. Act. represents the activation ratio of the feature dimension.

| Method | CIFAR-10 | | | | | CIFAR-100 | | | | | ImageNet-100 | | | | |
|---|---|---|---|---|---|---|---|---|---|---|---|---|---|---|---|
| | Cons. $\uparrow$ | $H_s \downarrow$ | $H_m \downarrow$ | $H_f \downarrow$ | Act. | Cons. $\uparrow$ | $H_s \downarrow$ | $H_m \downarrow$ | $H_f \downarrow$ | Act. | Cons. $\uparrow$ | $H_s \downarrow$ | $H_m \downarrow$ | $H_f \downarrow$ | Act. |
| Soft Gating | 51.84 | 1.09 | 1.12 | 1.40 | 0.78 | 8.54 | 3.32 | 3.34 | 3.86 | 0.70 | 33.15 | **2.03** | 2.76 | **2.26** | 0.48 |
| w/o Grad. Detach | 56.22 | 1.01 | 1.14 | 1.27 | 0.87 | 20.17 | 2.79 | **3.14** | 3.18 | 0.82 | 22.29 | 2.66 | 2.69 | 2.96 | 0.30 |
| BayesNCL | **56.50** | **0.99** | **1.12** | **1.25** | 0.89 | **22.02** | **2.70** | 3.20 | **3.09** | 0.87 | **36.14** | 2.10 | **2.44** | 2.37 | 0.50 |

*Table 10.* Comparing the Impact of the Complexities of Gating Head. $\uparrow$ indicates higher is better, $\downarrow$ indicates lower is better. Act. represents the activation ratio of the feature dimension.

| Method | CIFAR-10 | | | | | CIFAR-100 | | | | | ImageNet-100 | | | | |
|---|---|---|---|---|---|---|---|---|---|---|---|---|---|---|---|
| | Cons. $\uparrow$ | $H_s \downarrow$ | $H_m \downarrow$ | $H_f \downarrow$ | Act. | Cons. $\uparrow$ | $H_s \downarrow$ | $H_m \downarrow$ | $H_f \downarrow$ | Act. | Cons. $\uparrow$ | $H_s \downarrow$ | $H_m \downarrow$ | $H_f \downarrow$ | Act. |
| 1-Layer | **56.97** | **0.98** | 1.15 | **1.22** | 0.78 | **26.70** | **2.41** | **2.89** | **2.75** | 0.81 | **40.85** | 2.13 | **2.37** | 2.38 | 0.35 |
| 2-Layer | 56.50 | 0.99 | **1.12** | 1.25 | 0.89 | 22.02 | 2.70 | 3.20 | 3.09 | 0.87 | 36.14 | **2.10** | 2.44 | **2.37** | 0.50 |
| 3-Layer | 55.26 | 1.03 | 1.13 | 1.29 | 0.88 | 17.88 | 2.92 | 3.30 | 3.33 | 0.89 | 26.37 | 2.59 | 2.72 | 2.83 | 0.14 |

## I. Detailed Implementation Settings

In this section, we elaborate on the specific hyperparameter configurations to ensure reproducibility.

**General Optimization Settings.** For CIFAR-10 and CIFAR-100, the models are trained for 200 epochs with a batch size of 256, and the learning rate for the backbone is set to 0.4. For the ImageNet-100 dataset, we train for 100 epochs with a batch size of 128, and the learning rate is set to 0.15.

**BayesNCL Specific Settings.** Regarding the proposed method, we apply the generated binary masks to every feature vector element-wise to filter out task-irrelevant information. The KL-divergence weight $\lambda$ is fixed at $3 \times 10^{-5}$ across all datasets.

*Table 11.* **Effect of feature dimensionality** $K$ **on retrieval performance.** We report Precision@$k$ for different datasets. For CIFAR-10, $(k_1, k_2) = (1, 3)$; for CIFAR-100 and ImageNet-100, $(k_1, k_2) = (5, 10)$. BayesNCL consistently outperforms NCL across various sparsity levels, demonstrating its robustness in selecting discriminative features while suppressing common background noise.

| Dataset | Selected Dimensions | NCL (Baseline) | | BayesNCL (Ours) | |
|---|---|---|---|---|---|
| | | **P@**$k_1$ | **P@**$k_2$ | **P@**$k_1$ | **P@**$k_2$ |
| CIFAR-10 ($k_1$=1, $k_2$=3) | K=16 | 32.71% | 31.21% | 38.50% | 34.99% |
| | K=32 | 46.72% | 42.14% | 49.51% | 45.75% |
| | K=48 | 52.75% | 49.50% | 59.42% | 55.50% |
| | K=64 | 61.67% | 57.85% | 63.59% | 61.23% |
| | K=80 | 66.63% | 64.39% | 68.85% | 66.43% |
| | K=96 | 70.73% | 69.14% | 71.94% | 69.90% |
| CIFAR-100 ($k_1$=5, $k_2$=10) | K=16 | 4.84% | 4.80% | 4.67% | 4.53% |
| | K=32 | 8.64% | 7.89% | 9.04% | 8.46% |
| | K=48 | 12.00% | 10.95% | 13.06% | 11.99% |
| | K=64 | 14.54% | 13.32% | 16.20% | 14.84% |
| | K=80 | 16.91% | 15.63% | 19.41% | 17.57% |
| | K=96 | 19.78% | 18.24% | 22.81% | 21.11% |
| ImageNet-100 ($k_1$=5, $k_2$=10) | K=224 | 11.68% | 10.21% | 11.92% | 10.40% |
| | K=256 | 12.64% | 11.04% | 13.13% | 11.32% |
| | K=288 | 13.38% | 11.86% | 13.51% | 11.90% |
| | K=320 | 14.13% | 12.34% | 14.39% | 12.56% |
| | K=352 | 14.75% | 12.85% | 14.92% | 13.04% |
| | K=384 | 15.34% | 13.23% | 15.49% | 13.43% |

*Table 12.* The impact of different dimensions of representation on interpretability

| Dataset | dim | CL | | | | NCL | | | | BayesNCL | | | |
|---|---|---|---|---|---|---|---|---|---|---|---|---|---|
| | | Cons. ↑ | $H_s \downarrow$ | $H_m \downarrow$ | $H_f \downarrow$ | Cons. ↑ | $H_s \downarrow$ | $H_m \downarrow$ | $H_f \downarrow$ | Cons. ↑ | $H_s \downarrow$ | $H_m \downarrow$ | $H_f \downarrow$ |
| CIFAR-10 | 64 | 10.00 | 2.29 | 2.29 | 2.30 | 41.98 | 1.32 | 1.32 | 1.64 | 42.96 | 1.32 | 1.33 | 1.62 |
| | 128 | 10.00 | 2.29 | 2.29 | 2.30 | 49.97 | 1.17 | 1.17 | 1.47 | 50.62 | 1.13 | 1.20 | 1.43 |
| | 256 | 10.00 | 2.29 | 2.29 | 2.30 | 53.82 | 1.09 | 1.09 | 1.38 | 56.50 | 0.99 | 1.12 | 1.25 |
| | 512 | 10.00 | 2.29 | 2.29 | 2.30 | 57.62 | 1.00 | 1.00 | 1.27 | 58.10 | 0.99 | 1.02 | 1.25 |
| | 1024 | 10.00 | 2.29 | 2.29 | 2.30 | 61.88 | 0.90 | 0.91 | 1.16 | 62.81 | 0.88 | 0.93 | 1.13 |
| CIFAR-100 | 64 | 1.00 | 4.57 | 4.57 | 4.61 | 6.17 | 3.62 | 3.62 | 4.04 | 7.79 | 3.55 | 3.55 | 3.98 |
| | 128 | 1.00 | 4.57 | 4.57 | 4.61 | 8.43 | 3.39 | 3.41 | 3.87 | 12.28 | 3.13 | 3.25 | 3.55 |
| | 256 | 1.00 | 4.57 | 4.57 | 4.61 | 9.91 | 3.29 | 3.30 | 3.77 | 22.02 | 2.70 | 3.20 | 3.09 |
| | 512 | 1.00 | 4.57 | 4.57 | 4.61 | 13.36 | 3.05 | 3.06 | 3.56 | 25.89 | 2.52 | 3.21 | 2.86 |
| | 1024 | 1.00 | 4.57 | 4.57 | 4.61 | 15.72 | 2.89 | 2.92 | 3.40 | 20.06 | 2.77 | 3.02 | 3.25 |
| ImageNet-100 | 256 | 1.00 | 4.59 | 4.59 | 4.61 | 6.73 | 3.74 | 3.78 | 4.00 | 16.15 | 3.30 | 3.33 | 3.57 |
| | 512 | 1.00 | 4.59 | 4.59 | 4.61 | 9.74 | 3.57 | 3.72 | 3.76 | 22.81 | 2.78 | 2.80 | 2.98 |
| | 1024 | 1.00 | 4.59 | 4.59 | 4.61 | 12.11 | 3.38 | 3.61 | 3.46 | 22.31 | 2.98 | 3.05 | 3.24 |
| | 2048 | 1.00 | 4.59 | 4.59 | 4.61 | 14.93 | 3.28 | 3.55 | 3.26 | 36.14 | 2.10 | 2.44 | 2.37 |
| | 4096 | 1.00 | 4.59 | 4.59 | 4.61 | 17.58 | 3.04 | 3.46 | 2.89 | 24.92 | 2.25 | 2.26 | 2.57 |

*Table 13.* Comparing the impact of different prior probabilities $\rho$ on performance on the cifar100 dataset

| $\rho$ | Cons | $H_s$ | $H_m$ | $H_f$ | Acc@1 | Acc@5 |
|---|---|---|---|---|---|---|
| 0.1 | 5.66 | 3.68 | 3.68 | 4.09 | 56.97 | 84.07 |
| 0.2 | 6.03 | 3.65 | 3.65 | 4.07 | 57.37 | 84.90 |
| 0.3 | 6.64 | 3.57 | 3.57 | 4.03 | 58.31 | 85.36 |
| 0.4 | 7.13 | 3.52 | 3.52 | 3.99 | 59.15 | 85.19 |
| 0.5 | 17.94 | 2.80 | 3.24 | 3.15 | 60.04 | 86.25 |
| 0.6 | 20.66 | 2.67 | 3.27 | 3.05 | 60.02 | 85.93 |
| 0.7 | 19.92 | 2.72 | 3.19 | 3.10 | 59.84 | 85.82 |
| 0.8 | 22.02 | 2.70 | 3.20 | 3.09 | 60.73 | 86.62 |
| 0.9 | 18.43 | 2.89 | 3.34 | 3.31 | 60.66 | 86.97 |

*Table 14.* Compare the impact of different KL divergence loss weights $\lambda$ on performance on cifar100

| $\lambda$ | Cons | $H_s$ | $H_m$ | $H_f$ | Acc@1 | Acc@5 |
|---|---|---|---|---|---|---|
| $1e-5$ | 19.89 | 2.80 | 3.27 | 3.17 | 59.74 | 85.84 |
| $2e-5$ | 22.49 | 2.70 | 3.20 | 3.08 | 60.20 | 86.21 |
| $3e-5$ | 22.02 | 2.70 | 3.20 | 3.09 | 60.73 | 86.62 |
| $4e-5$ | 18.52 | 2.88 | 3.30 | 3.30 | 61.11 | 86.46 |
| $5e-5$ | 14.64 | 3.09 | 3.33 | 3.53 | 60.88 | 86.43 |
| $6e-5$ | 12.07 | 3.21 | 3.33 | 3.67 | 60.94 | 86.28 |
| $7e-5$ | 10.52 | 3.26 | 3.33 | 3.73 | 60.66 | 86.64 |
| $8e-5$ | 11.06 | 3.24 | 3.29 | 3.72 | 61.61 | 87.03 |
| $9e-5$ | 11.36 | 3.23 | 3.29 | 3.71 | 61.53 | 86.64 |

*Table 15.* Sensitivity analysis of the gating-head learning rate on CIFAR-100.

| LR | Cons. ↑ | $H_s$ ↓ | $H_m$ ↓ | $H_f$ ↓ | Acc@1 | Acc@5 |
|---|---|---|---|---|---|---|
| 0.04 | 15.75 | 3.01 | 3.27 | 3.43 | **60.92** | 86.45 |
| 0.10 | 22.02 | 2.70 | 3.20 | 3.09 | 60.69 | **86.62** |
| 0.20 | 20.83 | 2.69 | **2.97** | 3.10 | 60.08 | 85.96 |
| 0.40 | **25.05** | **2.51** | 3.02 | **2.83** | 60.27 | 86.46 |

The sparsity prior $\rho$ is set to 0.8 for CIFAR-10/100 and 0.6 for ImageNet-100. Furthermore, to stabilize the probabilistic estimation, the learning rate of the gating head is scaled by $0.25\times$ relative to the backbone's learning rate.

## J. Visualization of High-Entropy Dimensions in NCL

To verify the compositionality gap, we analyzed the feature space of a pre-trained NCL model. We calculated the entropy of each feature dimension across the validation set to identify "high-prevalence" features. Figures 5 through 7 visualize the images that maximally activate the top high-entropy dimensions.

**Observation:** These dimensions consistently capture non-semantic background textures, for example, grass and water, rather than object semantics. This confirms that deterministic similarity measures used by NCL entangles context with content, causing the model to incorrectly align samples based on background similarity. BayesNCL aims to suppress these nuisance features via probabilistic gating.

## K. Visualization of Gated Dimensions in BayesNCL

To validate that BayesNCL effectively resolves the optimization conflict inherent in deterministic nature, we analyze the semantic content of feature dimensions that are frequently suppressed by the gating mechanism. We identify dimensions where the gating probability is consistently high across the validation set. We then retrieve images that maximally activate these specific dimensions in the gated latent space $z$. Figures 8 through 10 visualize the top activated samples for three such high-suppression dimensions.

**Observation:** These images always contain some similar features. For example, Figure 9 isolates geometric grid patterns, such as cages, window screens, and keyboards. These results confirm our hypothesis regarding the compositionality gap. In the standard NCL framework, models can be very confused in these dimensions. By probabilistically gating these nuisance factors, BayesNCL prevents spurious alignment, ensuring that the contrastive loss focuses on the object semantics rather than contextual confounders, resulting in more consistent representations.

## L. Diminishing Utility at Scale with Large Batches

Inspired by an insightful question raised by an anonymous reviewer during the review process, we further investigate a totally intuitive hypothesis: *Since the optimization conflict manifests as gradient variance, can simply scaling up the batch size smooth out this variance and solve the problem for standard NCL?*

To definitively answer this, we conduct experiments on CIFAR-100 by progressively scaling the batch size for standard NCL and comparing it with our BayesNCL. The impact of batch size scaling on Semantic Consistency is reported in Table 16.

*Table 16.* The impact of batch size scaling on Semantic Consistency (CIFAR-100). BayesNCL remains robust, while NCL shows diminishing returns.

| Method | BS=256 | BS=512 | BS=1024 | BS=2048 | BS=4096 |
|---|---|---|---|---|---|
| NCL | 9.91 | 10.05 | 16.88 | 14.27 | 19.99 |
| BayesNCL (Ours) | **22.02** | **19.90** | **23.49** | **22.75** | **22.25** |

Indeed, we find that larger batches do smooth NCL's gradient variance, leading to a modest improvement in semantic consistency. However, we argue that this brute-force scaling is not a "free lunch" and cannot replace the fundamental mechanism of BayesNCL, for the following critical reasons:

**1. Prohibitive Computational Cost.** Scaling the batch size leads to a linear explosion in computational complexity. As shown in Table 17, a single forward pass of NCL at BS=4096 requires a massive 5.801 TFLOPs. In stark contrast, BayesNCL achieves superior consistency at the standard BS=256 setting with only 363.346 GFLOPs. Our probabilistic gating is orders of magnitude more efficient than simply averaging noise via large batches.

**2. Degradation of Downstream Generalization.** It is a well-documented phenomenon that extreme batch sizes can cause the optimizer to converge into sharp minima, thereby reducing the model's generalization ability. To verify this, we trained a linear probe using the NCL model pre-trained with BS=4096. As shown in Table 17, the downstream accuracy of NCL

(BS=4096) suffers a significant drop. Conversely, BayesNCL avoids these optimization pitfalls, maintaining strictly superior representation quality.

*Table 17.* The Cost-Performance Trade-off. Comparing the "Brute-force" NCL scaling against the standard BayesNCL on CIFAR-100.

| Setting | FLOPs (per forward pass) | Acc@1 (%) | Acc@5 (%) |
| --- | --- | --- | --- |
| NCL (BS=4096) | 5.801T | 55.03 | 82.70 |
| BayesNCL (BS=256) | **363.346G** | **60.69** | **86.62** |

**Conclusion.** We believe that the optimization conflict in NCL is inherent to its deterministic objective. Scaling the batch size merely averages the noise generated by a flawed objective. By contrast, BayesNCL structurally resolves the root cause of the conflict via probabilistic feature gating. The results confirm that our method is a fundamental and irreplaceable solution, rather than just a computationally cheap substitute for large batches.

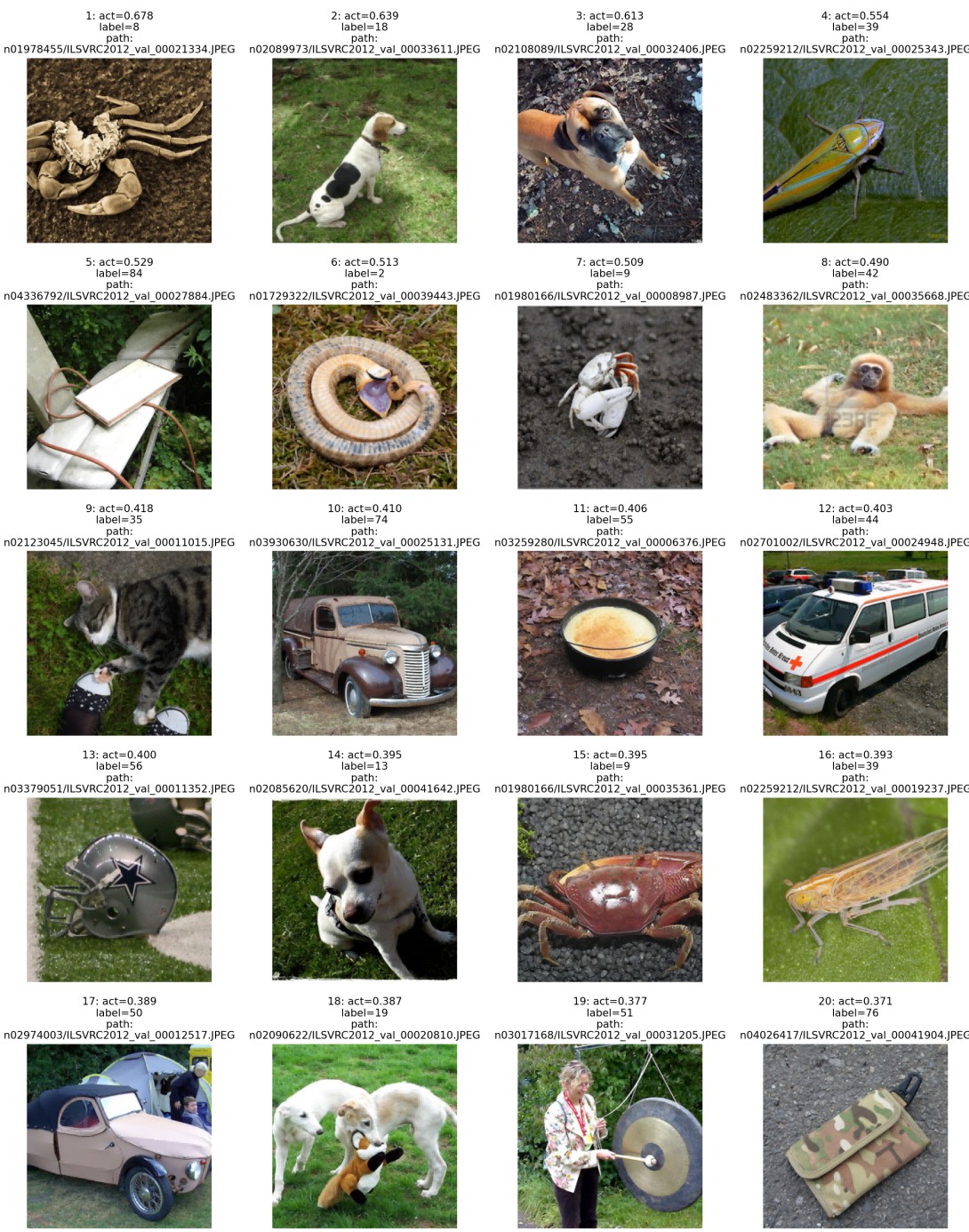

*Figure 5.* **NCL Dimension 1694.** This dimension activates strongly on green/grassy terrain, regardless of whether the foreground object is a dog or an insect. In NCL, this feature causes semantically distinct classes to be tightly clustered simply because they share a similar environment, illustrating a failure to disentangle content from context.

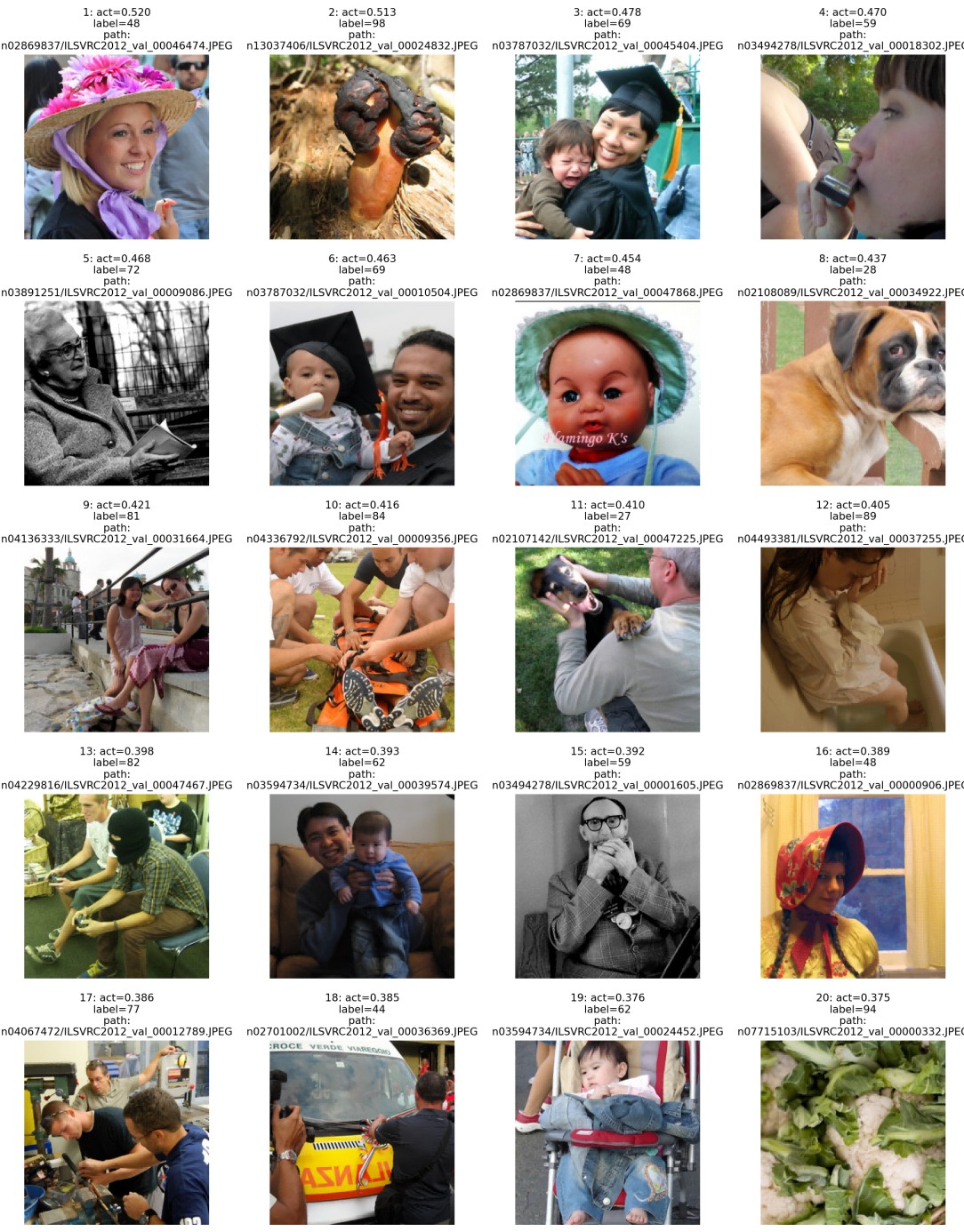

*Figure 6.* **NCL Dimension 261.** This dimension captures human faces and bodies. In many ImageNet classes (e.g., musical instruments, tools), a person is present but is not the label. NCL incorrectly learns this as a dominant feature, causing the model to align images based on the presence of a human rather than the actual target object.

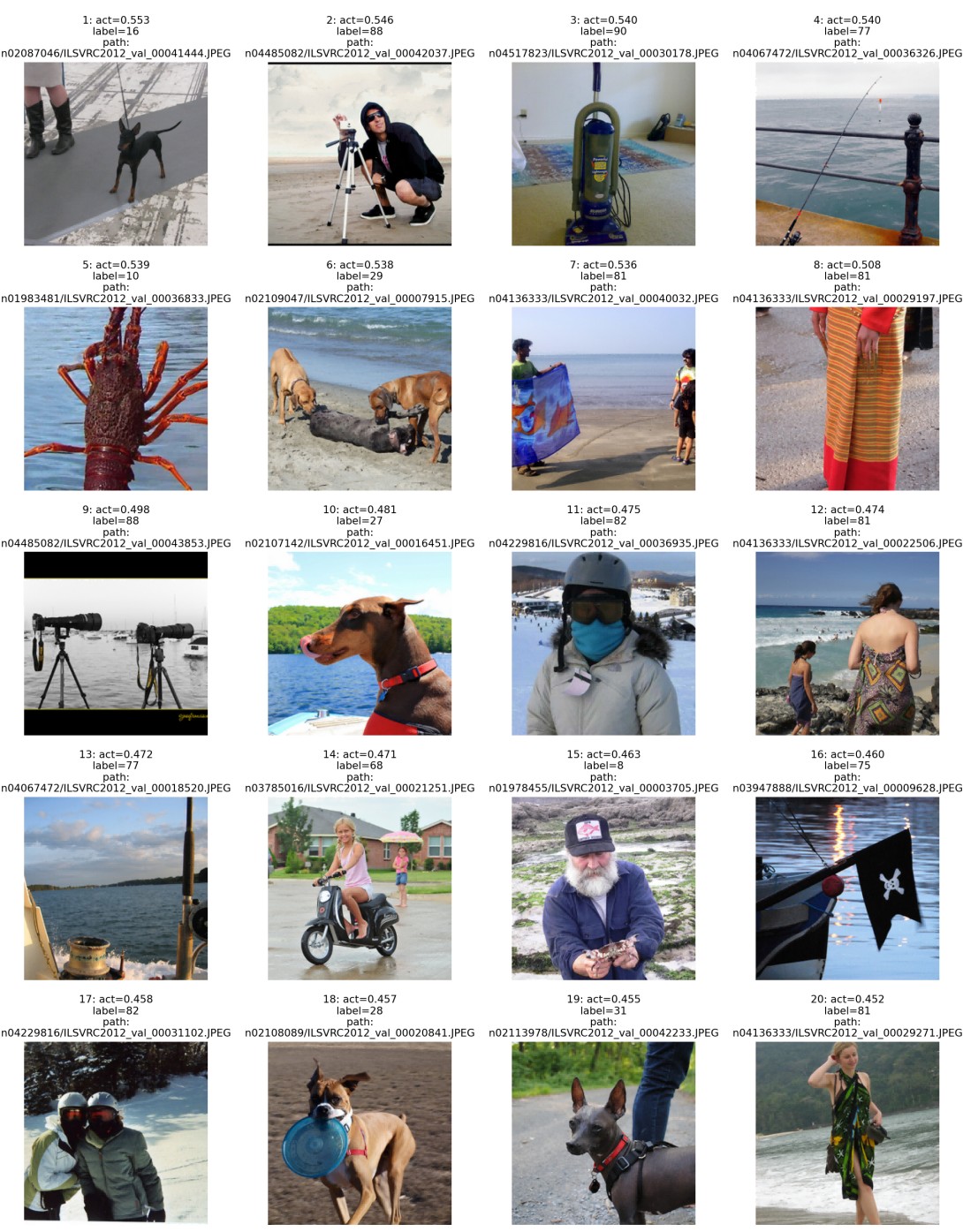

*Figure 7.* **NCL Dimension 1631.** This dimension correlates strongly with **blue backgrounds** (water, sky, snow). This creates a "compositionality gap": a boat on water and a boat on land will be pushed apart in the embedding space because this dominant background feature is inconsistent, violating semantic consistency.

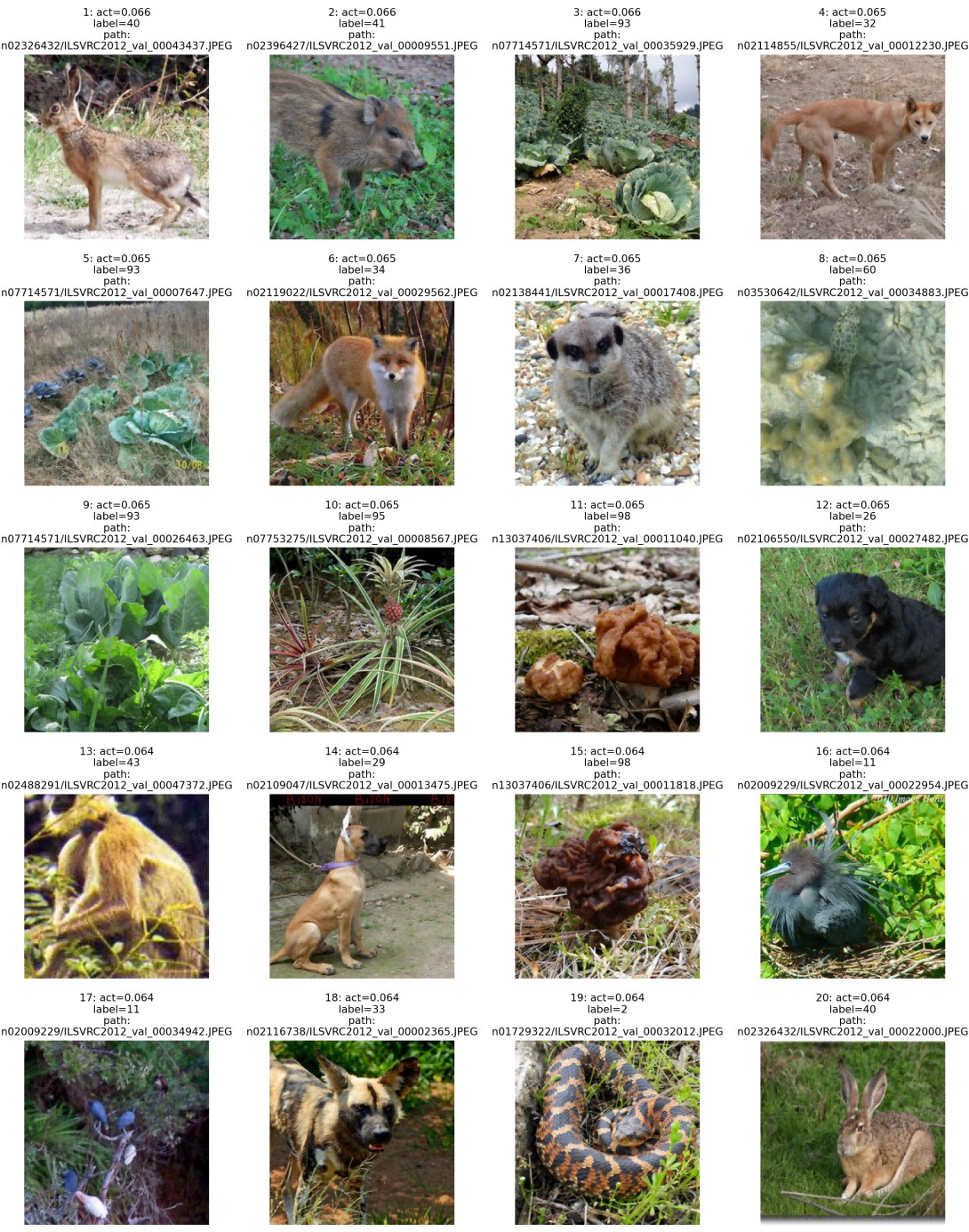

*Figure 8.* **Gated Dimension 1408.** This dimension is highly activated by foliage and grass textures. Note that the foreground objects vary widely (wildlife, fungi, domestic pets), yet the background remains consistent. BayesNCL learns to gate this dimension to prevent the model from clustering diverse classes based solely on their environment.

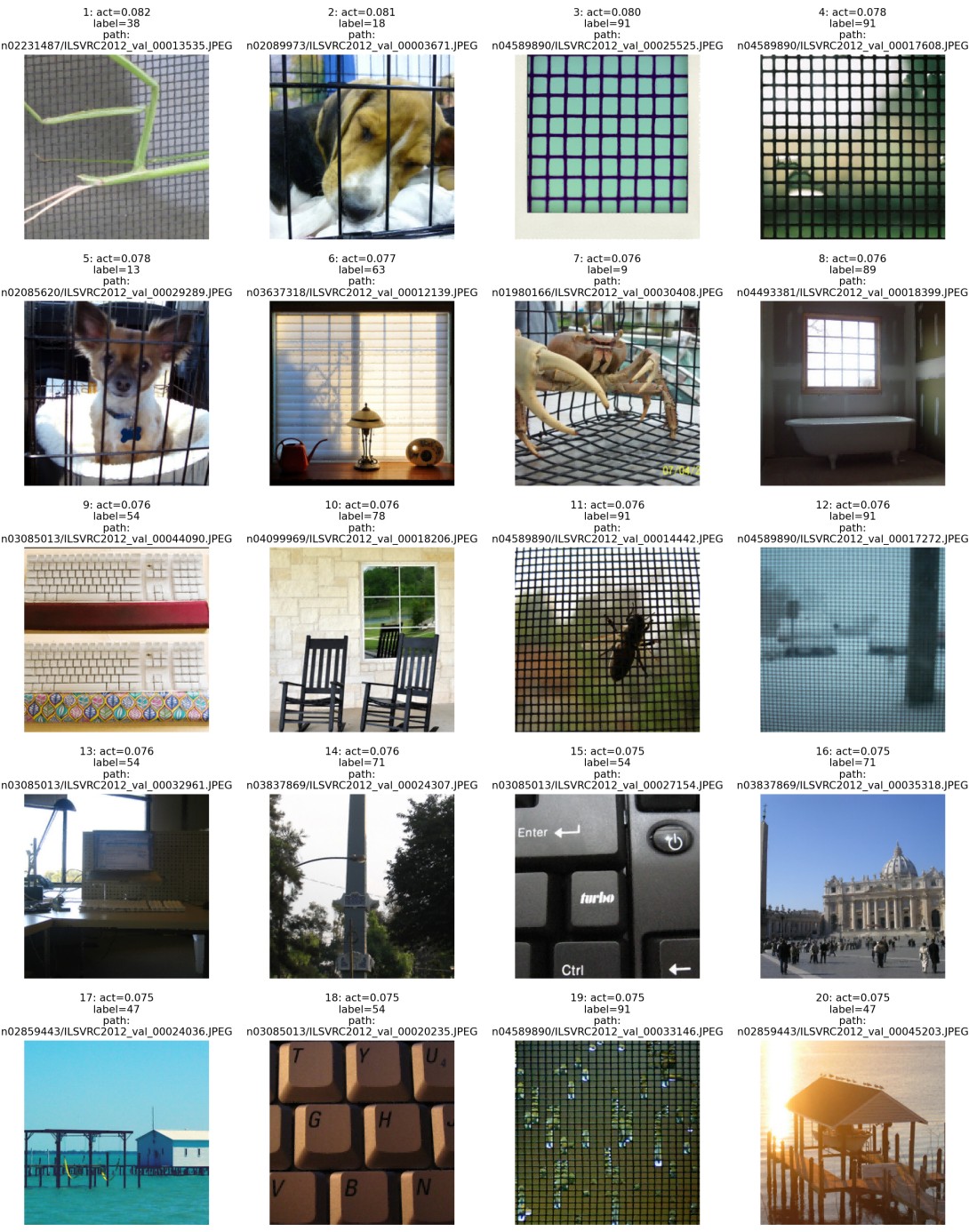

*Figure 9.* **Gated Dimension 1874.** This dimension captures grid-like patterns, including wire fences, window blinds, and computer keyboards. These are high-frequency texture features that carry little class-specific semantic information. By suppressing this dimension, BayesNCL avoids entangling "caged animals" with "office equipment."

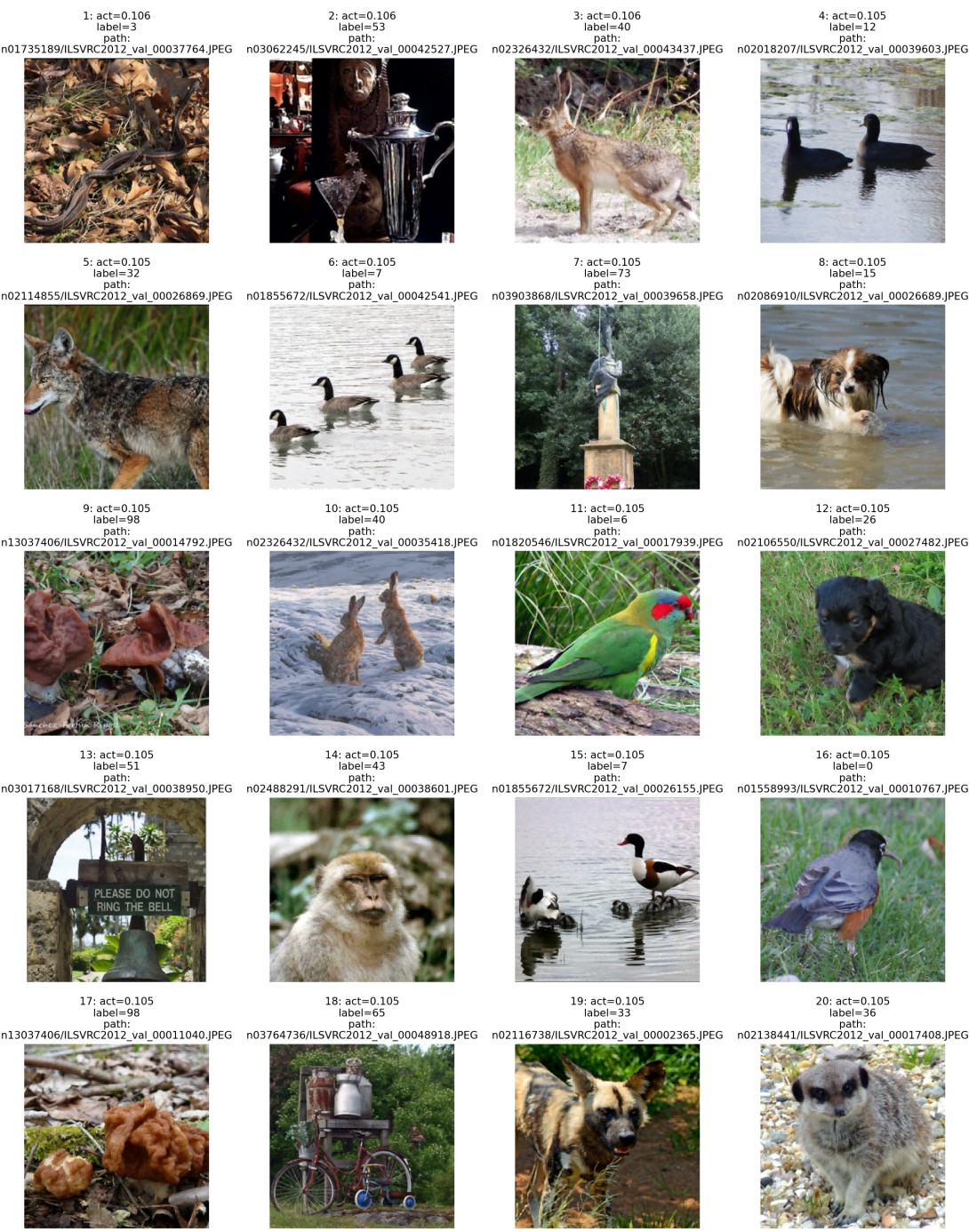

*Figure 10.* **Gated Dimension 1270.** The images activating this dimension are dominated by water, mud, and riverbanks. The model identifies this as a nuisance feature common to waterfowl, aquatic mammals, and landscapes, gating it to prioritize the distinguishing features of the objects themselves.

