# OpenReview forum: "Bayesian Gated Non-Negative Contrastive Learning"
_ICML.cc/2026/Conference — ICML 2026 regular_

### Official Review · Reviewer_PbsT · 2026-03-06

**Soundness:** 3
**Presentation:** 2
**Significance:** 3
**Originality:** 2
**Overall Recommendation:** 4
**Confidence:** 3

**Summary:**

This paper points out a key defect in the current non-negative contrastive learning (NCL) methods: the "optimization conflict" caused by the deterministic similarity measure when dealing with composite scenes with shared background features (such as "blue sky"). Positive samples pairs will encourage these shared features to remain consistent, while negative sample pairs will simultaneously pull them apart, resulting in gradient oscillations, which hinders precise semantic separation. To solve this problem, the authors proposed BayesNCL (Bayesian gated non-negative contrastive learning), which introduces a probabilistic gating mechanism to dynamically filter out irrelevant and high-frequency shared features related to the task. By formalizing the feature selection as a variational inference problem with sparse Bernoulli priors and optimizing it through a direct estimator (STE), BayesNCL successfully resolves the gradient conflict. Experiments show that compared with the state-of-the-art baseline, BayesNCL achieves a significant 142.1% relative improvement in semantic consistency on ImageNet-100, while not sacrificing downstream representational capabilities.

**Compliance With Llm Reviewing Policy:**

Affirmed.

**Final Justification:**

Most of my concerns have been addressed well. I keep my original rating.

**Key Questions For Authors:**

(i) Please provide a detailed discuss to explain why the assumption used in Appendix E.2 is reasonable and acceptable.

(ii) Please provide a quantitative analysis for the computational cost of the proposed BayesNCL.

(iii) Please provide an ablation analysis or discuss to justify why the specific ratio of learning rate scaling is 0.25.

(iv) There are some grammatical errors, format errors and notation overloading in the article. Please further polish and revise them.

**Limitations:**

The proposed method has two main limitations. Firstly, the introduction of the probabilistic gated head and skip connection estimator (STE) naturally leads to an increase in computational cost during the training process. Secondly, although the 2048-dimensional representation is more efficient than the 4096-dimensional parameters required by the NCL baseline, relying on such a high-dimensional latent space is still extremely large compared to the standard representation learning setup.

**Strengths And Weaknesses:**

Strength

(i) This paper focuses on a key and significant bottleneck issue in interpretability of contrastive learning: the optimization conflicts caused by common background features.

(ii) The proposed BayesNCL dynamically filters out high-frequency noise using a sparse Bernoulli prior by transitioning from deterministic similarity to a probabilistic gating mechanism. This method approximates Inverse Prevalence Weighting, offering a novel structural information bottleneck to disentangle semantics without requiring prior knowledge of the data distribution.

(iii) The paper provides a comprehensive empirical evaluation to explain the performance of BayesNCL. BayesNCL has made significant progress in interpretability metrics and performs equally well or even slightly better than baseline models such as SimCLR and NCL in maintaining or even slightly surpassing the linear probe accuracy.

Weakness

(i) The proof of Theorem 4.7 (Appendix E.2) seems to rely on a strong assumption that the original, pre-gated features ($z$) remain consistent in both deterministic NCL and Bayesian NCL. However, introducing gated heads and sparse losses during the end-to-end training process may fundamentally alter the learning trajectory of the encoder, meaning that the base representation $z_{Bayesian}$ may differ from $z_{NCL}$. The paper does not provide sufficient explanations to justify why this assumption is reasonable and acceptable.

(ii) This paper does not provide a quantitative analysis for the computational cost of the proposed BayesNCL. BayesNCL introduces a Gating Head to predict Bernoulli parameters and relies on the Straight-Through Estimator (STE) during training, which may result in additional computational costs.

(iii) The learning rate of the gating head is scaled by 0.25 relative to the backbone in Appendix I, but no ablation or discuss is provided to justify this specific ratio.

(iv) Has some loopholes, e.g.,

- Grammatical errors, e.g., "Comparation of the effects of different..." in Table 3 seems to be "Comparison" rather than "Comparation".
- Format errors, e.g., Eq.1&8 extend beyond the column margins.
- Notation overloading, e.g., $\tilde{z}$ is defined as Gated Feature Representation ($\tilde{z} = z \odot m_{train}$) in Table 5, but $\bar{z}_{j,c}$ is used as the average activation value of the $c$-th class.

Please correct the mistakes and polish them if possible.

---

> ### Author Rebuttal · Authors · 2026-03-30
>
> Thank you for your positive and thoughtful feedback. Please find our response below:
> # 1. [W1, Q1]
> We thank the reviewer for this insightful question. We would like to clarify that anchoring the base feature $z$ in Theorem 4.7 is a deliberate and mathematically sound assumption. Specifically, assuming $z$ remains consistent across both methods for this theoretical analysis is reasonable and necessary for the following three reasons:
> 1. To rigorously isolate the pure effect of the Bayesian gating mechanism, we must perform a controlled comparison at an identical state. If we evaluated the two objectives using different features, i.e. $z_{NCL}$ and $z_{Bayes}$, the variables would be confounded.
> 2. Theorem 4.7 is fundamentally designed to analyze the local loss landscape at any arbitrary training step $t$, rather than the final converged state. By anchoring the same current feature $z_t$, the theorem proves that at any given coordinate in the latent space, BayesNCL consistently provides a cleaner contrastive signal (with fewer background-induced false positive errors) compared to standard NCL.
> 3. This local assumption is exactly what explains the superior global representation. Precisely because BayesNCL computes a more accurate local gradient at every single step (which can only be proven by anchoring $z$), the continuous accumulation of these corrected gradients is what fundamentally steers the encoder toward the highly disentangled final representation we observe empirically.
>
> In summary, anchoring $z$ is a standard and necessary theoretical practice to evaluate the step-wise effectiveness of the algorithmic modification. We will explicitly add this clarification to Appendix E.2 to ensure the scope and intent of this assumption are perfectly clear.
>
> # 2. [W2, Q2, L1] Computational efficiency
> For each method, we conducted three experiments to evaluate the average running time and recorded the computational complexity of the models during training. The results on the Cifar100 are as follows:
>
> | Method   | Running Time (min) | FLOPs  |
> | :------- | :----------------- | :----- |
> | NCL      | 70.95              | 1.416G |
> | BayesNCL | 75.12              | 1.419G |
>
> The results on the Imagenet100 are as follows:
>
> | Method   | Running Time (min) | FLOPs  |
> | :------- | :----------------- | :----- |
> | NCL      | 193.78             | 3.731G |
> | BayesNCL | 218.53             | 3.815G |
>
> As shown in the table above, the introduction of the gated head and STE mechanism only incurs a small computational overhead. Given the significant improvement in semantic consistency, we believe the additional resource requirement is acceptable.
>
> # 3. [W3, Q3] Ablation experiments on gated head learning rates
> The table below provides our ablation experiments on learning rate of gating head on cifar100. Experiments show that 0.1 (i.e. 0.25 times the backbone network learning rate) is the most appropriate choice.
>
> | lr   | Cons.      | $H_s$    | $H_m$    | $H_f$    | Acc@1     | Acc@5     |
> | ---- | --------- | -------- | -------- | -------- | --------- | --------- |
> | 0.04 | 15.75     | 3.01     | 3.27     | 3.43     | **60.92** | 86.45     |
> | 0.1  | 22.02     | 2.70     | 3.20     | 3.09     | 60.69     | **86.62** |
> | 0.2  | 20.83     | 2.69     | **2.97** | 3.10     | 60.08     | 85.96     |
> | 0.4  | **25.05** | **2.51** | 3.02     | **2.83** | 60.27     | 86.46     |
>
> # 4. [W4, Q4] Grammatic Typos
> Thank you very much for your suggestions, we will refine them in the final version.
>
> # 5. [L2] Dimension requirements
> We thank the reviewer for this comment. First, we would like to briefly clarify that the original NCL baseline actually adopts a 2048-dimensional representation, rather than 4096. Our choice of 2048 dimensions in the main experiments was strictly to ensure a fair comparison under the exact same settings. Regarding the representation size, we fully agree that evaluating under standard lower-dimensional setups is important. To address this, Figure 4a and Table 9 explicitly present the interpretability metrics evaluated under various output dimensions. The results clearly demonstrate that our method consistently outperforms the NCL baseline across all tested dimensionalities. This indicates that our advantages in interpretability are robust and hold strong regardless of the latent space size, without relying on an unusually large dimension.

---

> > ### Author Rebuttal · Reviewer_PbsT · 2026-03-31
> >
> > No other concerns

---

> > > ### Author Response · Authors · 2026-04-04
> > >
> > > Dear Reviewer PbsT,
> > >
> > > Thank you for supporting the acceptance of this paper. We're glad that our responses addressed your concerns. We truly appreciate your valuable time for the reviewing.
> > >
> > > Best regards,
> > >
> > > Authors of submission 4516

---

### Official Review · Reviewer_9W98 · 2026-03-12

**Soundness:** 3
**Presentation:** 3
**Significance:** 3
**Originality:** 3
**Overall Recommendation:** 5
**Confidence:** 4

**Summary:**

This paper proposes the BayesNCL, a contrastive learning approach for learning disentangled features. The authors identify and formalize the phenomenon of "optimization conflict" caused by common but task-irrelevant features, observed in the original non-negative contrastive learning (NCL). To address this problem, the authors propose a Bayesian Gating mechanism that allows the model to dynamically filter out task-irrelevant common features. The effectiveness of the proposed method is demonstrated by both theoretical analysis and empirical comparisons.

**Compliance With Llm Reviewing Policy:**

Affirmed.

**Final Justification:**

The rebuttal solved my concerns, so I lean towards acceptance.

**Key Questions For Authors:**

1. Will the gating mechanism harm the learning of common features, e.g., retrieval of images with certain backgrounds?
2. Section D.2, Eq (21). Should it be $-\mathcal{L}\_{reconstruction}$?

**Limitations:**

Yes.

**Strengths And Weaknesses:**

**Strength**
1. The paper is well written and easy to follow.
2. The proposed method is supported by statistical theory and gradient analysis.
3. The experimental results are thorough and strong.

**Weakness**
1. The tables lie too far away from their discussions, e.g., Table 2 is on top page 4, but has not been referred to until the end of page 7. This hinders readability.
2. The paper uses background as an example of common but task-irrelevant features. However, during pretraining, we typically do not know the downstream tasks, so we are not supposed to assume if a feature is task-relevant or not. I think the authors should discuss whether this gating approach harms the learning of common features, as they can be task-relevant too.
3. (Minor) The 142.1% improvement is a bit overclaimed in the abstract. It should be noted that it is on Imagenet100. The performance gain on, e.g. Cifar10, is not that significant.

---

> ### Author Rebuttal · Authors · 2026-03-30
>
> Thank you for your recognition and valuable suggestions. Please find our response below:
> # 1. [W1, W3] Layout and presentation issues
> Thank you very much for your suggestions, we will carefully incorporate them into the final version to ensure a more coherent narrative and eliminate any potential ambiguities.
>
> # 2. [W2, Q1] Will the gating mechanism harm the learning of common features?
> To directly verify whether our method harms common features like backgrounds, we conducted a background classification experiment.
>
> Specifically, we used the frozen backbones pre-trained on ImageNet-100 (via NCL and BayesNCL) to train a single-layer linear classification head on the Waterbirds dataset, explicitly predicting the background labels. We report the mean of five random seeds.
>
> | Method   | Acc (%) |
> | :------- | :------ |
> | NCL      | 93.33   |
> | BayesNCL | 93.38   |
>
> If BayesNCL completely drops or harms background features, then the accuracy on this task should be significantly lower than NCL. However, we find that BayesNCL maintains (and even slightly improves) the background classification accuracy compared to the NCL baseline. This serves as direct evidence that BayesNCL does not discard or harm coherent background structures, but rather effectively preserves them while mitigating foreground-background entanglement.
> # 3. [Q2] Symbol problem
> Yes, you are absolutely correct, and we appreciate your careful reading. This was simply a typo in our original manuscript. In Eq. (20), the term $E_{m \sim q_\theta}[\log p(z^+|z, m)]$ represents the expected log-likelihood, which we aim to maximize. However, following standard conventions, the symbol $L$ denotes a "Loss" that should be minimized. Therefore, to correctly represent the ELBO using loss notation, Eq. (21) should indeed be written as $-L_{reconstruction} - L_{regularization}$. We will fix this sign error and ensure the notation strictly aligns with standard loss conventions in the final version.

---

> > ### Author Rebuttal · Reviewer_9W98 · 2026-04-03
> >
> > My concerns have been addressed. I will keep my score.

---

> > > ### Author Response · Authors · 2026-04-04
> > >
> > > Dear Reviewer 9W98,
> > >
> > > Thank you for supporting the acceptance of this work. We truly appreciate your valuable reviews and suggestions.
> > >
> > > Best regards,
> > >
> > > Authors of submission 4516

---

### Official Review · Reviewer_Q8qy · 2026-03-13

**Soundness:** 3
**Presentation:** 3
**Significance:** 3
**Originality:** 3
**Overall Recommendation:** 4
**Confidence:** 3

**Summary:**

This paper introduces a mechanism to reduce "optimization conflict" in contrastive learning (CL). Optimization conflict refers to certain features potentially contributing to both the positive and negative terms of the CL objective, hence leading to high variance gradients. The proposed solution is to probabilistically gate feature dimensions based on their frequency, so that common features, which are shared between the positive and negative terms, will be downweighted. This fix leads to better disentangled representations that map discriinative features to separate dimensions. This is achieved while maintaining the representation's downstream performance quality.

**Compliance With Llm Reviewing Policy:**

Affirmed.

**Final Justification:**

I was positive on this paper in my first review and the rebuttal has only strengthened things. I think the batch size experiments are a particularly nice addition. I'm sticking with a score of 4 to indicate that I think this paper should be accepted but that there are other papers that I found stronger, for which I reserved higher scores.

**Key Questions For Authors:**

I'm curious if you see BayesNCL as critical for resolving the "optimization conflict," or if that conflict goes away for the baselines given sufficient scale, hyperparams, etc. My guess is the latter. First, gradient variance can be reduced with bigger batches; better optimizers might also suffer less from this variance. Second, the InfoNCE objective already has the property that common features will be downweighted, in the following sense: the minimizer of the objective is $s(z, z^{\prime}) = \texttt{PMI}(z, z^{\prime}) + C = \log \frac{p(z,z^{\prime})}{p(z)p(z^{\prime})} + C$. The denominator therefore already downweights common features. So it should downweight sky, no? Is $s_{IPW}$  doing something importantly different, that goes beyond what standard InfoNCE will already do in theory? Or is it more that $s_{IPW}$ helps get there faster or applies stronger regularization toward the same effect? In any case, I think the paper would benefit from some thought about this.

**Limitations:**

There isn't much discussion of limitations or associated costs of BayesNCL. What are the tradeoffs involved in using it versus NCL or other baselines? It seems to me like there could be a few: more complicated, extra compute, optimization difficulties with STE. And what are the limitations in scope of applicability? Does it work for settings where the data is not well explained by a set of latent classes?

**Strengths And Weaknesses:**

Strengths:
1. The theoretical motivation is mostly compelling (but I do have some questions about it below)
2. The method is novel and seems like a reasonable solution
3. Performance gains are substantial
4. The experiments and theoretical analysis are both fairly extensive

Weaknesses:
1. Some statements, especially early in paper, are made without immediate justification. Some of these are later justified and others are not. The writing could be tightened to avoid overly broad claims, and forward references could be added to claims that will later be justified. Here are some examples I noticed:
* "In typical contrastive frameworks, the learned embedding space is highly entangled, where semantic concepts are distributed holistically across all feature dimensions rather than being aligned with specific axes." [citation needed]
* "This optimization conflict induces severe gradient oscillation on shared dimensions, effectively obstructing the emergence of the clean, disentangled representations that NCL inherently seeks to achieve.” [citation needed, or forward reference to the gradient variance analysis]
* "This conflict results in gradient oscillation and hinders semantic consistency." [where is the direct evidence of hindering semantic consistency?]
* "This prevents the optimizer from settling into a sparse solution." [the preceding proposition does not directly show this]

2. The first half of the paper could benefit from analytical experiments that validate the theory. For example, I would like to see some quantification of the degree of gradient oscillation in standard CL, and some evidence that this is causally related to failures of semantic consistency. The experiments only come in Section 5, and they mainly come across as benchmarking the method rather than checking the theory.

Minor:
1. "probabilitically" --> "probabilistically"

---

> ### Author Rebuttal · Authors · 2026-03-30
>
> Thank you for your positive and constructive feedback. Please find our response below:
> # 1. [W1] Broad expressions
> Thank you very much for your suggestions, we will refine them in the final version to ensure the description is precise and to avoid any potential ambiguity.
> # 2. [W2] Evidence for gradient oscillations
> To empirically validate "Gradient Instability," we analyzed gradient dynamics during NCL pre-training on CIFAR-100. We computed Spearman correlations among Activation Frequency (AF), Gradient Variance (GV), and Semantic Consistency (SC) per dimension every 10 epochs.
>
> | epoch | AF-GV | AF-SC   | GV-SC  |
> | ----- | ----- | ------- | ------ |
> | 10    | 0.748 | -0.880  | -0.646 |
> | 20    | 0.753 | -0.861  | -0.675 |
> | 30    | 0.787 | -0.906  | -0.688 |
> | 40    | 0.728 | -0.869  | -0.627 |
> | 50    | 0.816 | -0.906, | -0.706 |
> | 60    | 0.705 | -0.912  | -0.596 |
> | 70    | 0.694 | -0.881  | -0.607 |
> | 80    | 0.703 | -0.882  | -0.605 |
> | 90    | 0.635 | -0.880  | -0.522 |
> | 100   | 0.735 | -0.910  | -0.643 |
>
> The strong positive AF-GV correlation confirms that prevalent features drive gradient instability. The persistent negative GV-SC correlation shows these oscillations hinder semantic disentanglement, directly validating our Optimization Conflict theory. Thanks for your suggestion, we will add this result to the final version to verify our theoretical analysis.
> # 3. [Q1] Impact of larger batch size
> We sincerely appreciate this insightful question. Honestly, we are also curious about this question. To definitively answer this, we conducted experiments on the CIFAR-100 with larger batch size.
>
> |          | 256   | 512   | 1024  | 2048  | 4096  |
> | -------- | ----- | ----- | ----- | ----- | ----- |
> | NCL      | 9.91  | 10.05 | 16.88 | 14.27 | 19.99 |
> | BayesNCL | 22.02 | 19.90 | 23.49 | 22.75 | 22.25 |
>
> As you intuited, larger batches smooth NCL's gradient variance. However, scaling up is not a free lunch and introduces well-known compromises like converging into sharp minima, which reduces downstream generalization ability and massive memory costs. Furthermore, bs=256 is the standard setting in the solo-learn [1] framework. More importantly, we believe that NCL's conflict is inherent to its deterministic objective; scaling merely averages the noise of a flawed objective. By contrast, BayesNCL fundamentally resolves the conflict via probabilistic feature gating. This proves our method is critical, not just a substitute for large batches.
> # 4. [Q2] Function of $S_{IPW}$
> We fully agree that the InfoNCE minimizer inherently down-weights common features. However, the critical distinction lies not in whether down-weighting occurs, but how it is physically realized during gradient descent.
>
> In standard InfoNCE (dot product), down-weighting common "sky" in negative pairs forces the network to suppress the "sky" feature activations via gradients. Since positive pairs demand maximizing these same activations, this causes the optimization conflict.
>
> BayesNCL goes beyond standard regularization by altering the metric space. Using $S_{IPW}$ and gating, it dynamically reduces the metric weight of "sky" rather than penalizing its feature activations. This allows the model to maintain stable activations for common features without gradient tug-of-wars, enabling true disentanglement.
> # 5. [L1] Computational efficiency
> For each method, we conducted three experiments to evaluate the average running time and recorded the computational complexity of the models during training. The results on the Cifar100 are as follows:
>
> | Method   | Running Time (min) | FLOPs  |
> | :------- | :----------------- | :----- |
> | NCL      | 70.95              | 1.416G |
> | BayesNCL | 75.12              | 1.419G |
>
> The results on the Imagenet100 are as follows:
>
> | Method   | Running Time (min) | FLOPs  |
> | :------- | :----------------- | :----- |
> | NCL      | 193.78             | 3.731G |
> | BayesNCL | 218.53             | 3.815G |
>
> As shown in the table above, the introduction of the gated head and STE mechanism only incurs a small computational overhead. Given the significant improvement in semantic consistency, we believe the additional resource requirement is acceptable.
> # 6. [L2] Scope of application
> BayesNCL assumes data is "compositional" and decomposable into discrete latent concepts. While typically true for real-world images, this assumption fails for data lacking clear latent boundaries, for example, highly entangled or non-compositional continuous signals. In such cases, our gating mechanism might struggle to identify meaningful semantic dimensions to mask. We will explicitly discuss this applicability boundary in the Limitations section of the final version.
>
> ---
>
> [1] Da Costa, V. G. T., Fini, E., Nabi, M., Sebe, N., & Ricci, E. (2022). solo-learn: A library of self-supervised methods for visual representation learning. _Journal of Machine Learning Research_, _23_(56), 1-6.

---

> > ### Author Rebuttal · Reviewer_Q8qy · 2026-04-03
> >
> > Thanks for the rebuttal!
> >
> > The gradient variance analysis is a nice addition.
> >
> > The batch size experiments are a great comparison. It does, however, show that some of the effect can be achieved just with bigger batches. Do the bigger batches have a cost in terms of increased training flops? It seems like it should... I would be interested to see if BayesNCL is more efficient compared to NCL with big batches.

---

> > > ### Author Response · Authors · 2026-04-04
> > >
> > > Thank you very much for your further reply. Please find our response below:
> > > # [Q1]
> > > We separately counted the FLOPs of NCL during a single forward pass with batch sizes of 256, 512, 1024, 2048, and 4096. The results are as follows:
> > >
> > > | Batch Size | 256      | 512      | 1024   | 2048   | 4096   |
> > > | ---------- | -------- | -------- | ------ | ------ | ------ |
> > > | FLOPs      | 362.541G | 725.082G | 1.450T | 2.900T | 5.801T |
> > >
> > > In contrast, BayesNCL's FLOPs for a single forward pass in the standard setting are as follows:
> > >
> > > | Method   | FLOPs    |
> > > | -------- | -------- |
> > > | BayesNCL | 363.346G |
> > >
> > > As can be seen, our method is more efficient than simply using larger batches of NCL. In addition, we trained a linear probe using a model pre-trained with NCL with a batch size of 4096, and report the mean of three random seeds.
> > >
> > > |               | Acc@1 | Acc@5 |
> > > | ------------- | ----- | ----- |
> > > | NCL (bt=4096) | 55.03 | 82.70 |
> > > | BayesNCL      | 60.69 | 86.62 |
> > >
> > > The results show that although increasing the batch size can smooth out the variance, it comes at the cost of downstream task performance. In contrast, BayesNCL is more efficient and maintains or even improves downstream task performance, making BayesNCL a better and irreplaceable choice than simply increasing the batch size for NCL.

---

### Official Review · Reviewer_VJNk · 2026-03-13

**Soundness:** 3
**Presentation:** 2
**Significance:** 3
**Originality:** 3
**Overall Recommendation:** 3
**Confidence:** 3

**Summary:**

The authors propose a novel feature bayesian gating mechanism as an approach to prevent feature conflicts during SSL training of non-negative contrastive learning methods. To justify their approach they provide examples of background-correlated feature dimensions, and mathematically analyze the conflict in the training objective. Their gating mechanism seeks to predict a sparse mask (determined by prior $\rho$) based on a Bernoulli random variable that seeks to intelligently “turn off” feature dimensions that have high prevalence. Their results suggest that their approach both learns better disentanglement over NCL and other contrastive methods in addition to a strong representation for linear probing. They provide additional experiments on their choices of gradient computation, architecture and hyperparameter selection.

**Compliance With Llm Reviewing Policy:**

Affirmed.

**Key Questions For Authors:**

- Is it exactly harmful to learn a coherent background feature? One of the apparent strengths of SSL is to learn the structure of the data (perhaps including background features) that can adapted easily to downstream tasks.
- What does this look like for other SSL methods that do not use negative pairs? Given there is no negative pairs, is there still any gradient instability, how does the disnentanglement look like for these methods?

**Limitations:**

Yes

**Strengths And Weaknesses:**

- Strengths
    - The method appears technically sound and seems to have the desired effect, improving the per-dimension class disentanglement in the learned representation.
    - The method performs well outperforming existing baselines on the same architecture and training setup.
- Weaknesses
    - It would be convincing to have an example of the gradient variance that occurs without the gating mechanism.
    - Table 1 is difficult to understand, baseline methods are not included and appear to only be referred to in text. Metric $H_f$ is also not defind in the metrics.
    - Table 4 is not referred to anywhere in the paper. It would also be helpful to provide
    - It is not entirely clear what the experimental setup is in Figure 3, and is not elaborated on when it is referred to in the **Representation Quality** section.

---

> ### Author Rebuttal · Authors · 2026-03-30
>
> Thank you for your insightful and detailed comments. Please find our response below:
> # 1. [W1] Evidence for gradient oscillations
> To empirically validate "Gradient Instability," we analyzed gradient dynamics during NCL pre-training on CIFAR-100. We computed Spearman correlations among Activation Frequency (AF), Gradient Variance (GV), and Semantic Consistency (SC) per dimension every 10 epochs.
>
> | epoch | AF-GV | AF-SC   | GV-SC  |
> | ----- | ----- | ------- | ------ |
> | 10    | 0.748 | -0.880  | -0.646 |
> | 20    | 0.753 | -0.861  | -0.675 |
> | 30    | 0.787 | -0.906  | -0.688 |
> | 40    | 0.728 | -0.869  | -0.627 |
> | 50    | 0.816 | -0.906, | -0.706 |
> | 60    | 0.705 | -0.912  | -0.596 |
> | 70    | 0.694 | -0.881  | -0.607 |
> | 80    | 0.703 | -0.882  | -0.605 |
> | 90    | 0.635 | -0.880  | -0.522 |
> | 100   | 0.735 | -0.910  | -0.643 |
>
> The strong positive AF-GV correlation confirms that prevalent features drive gradient instability. The persistent negative GV-SC correlation shows these oscillations hinder semantic disentanglement, directly validating our Optimization Conflict theory.
> # 2. [W2] Table 1 is difficult to understand
> We will provide a detailed introduction of each baseline in the appendix section and mention them in the header of Table 1. The strict mathematical definitions of interpretability metrics are provided in Appendix F, and we will also mention them in the header.
> # 3. [W3] Table 4 is not referred
> Thank you very much for raising this question. We will provide a relevant explanation in Section 5.2 of the main text.
> # 4. [W4] Experimental setup in Figure 3
> We use models pre-trained on NCL and BayesNCL respectively for feature selection, and select the Top-k dimensions with the largest activation values for image retrieval.
> # 5. [Q1] Is learning background features harmful?
> We want to clarify that background features themselves are not "harmful"; rather, it is their entanglement leads to performance degradation. As we analyzed in the paper, separating negative sample pairs with the same background leads to gradient conflicts in specific dimensions, which destroys these features. BayesNCL addresses this issue by acting as a feature decoupler, not a deleter. Our empirical results also confirm this:
>
> 1. Table 1 shows that BayesNCL maintains or increases the activation rate (Act.), indicating the model utilizes more independent dimensions to capture features rather than discarding them.
> 2. Linear probing performance in Table 2 shows that BayesNCL improves linear separability without sacrificing information.
> 3. To directly verify whether our method harms common features like backgrounds, we conducted a background classification experiment on the Waterbirds dataset. Specifically, we used the frozen backbones pre-trained on ImageNet-100 (via NCL and BayesNCL) to train a single-layer linear classification head, explicitly predicting the background labels. We report the mean of five random seeds.
>
> | Method   | Acc   |
> | :------- | :---- |
> | NCL      | 93.33 |
> | BayesNCL | 93.38 |
>
> If BayesNCL harms background features, then the accuracy on this task should be significantly lower than NCL. However, we find that BayesNCL maintains (and even slightly improves) the background classification accuracy compared to the NCL baseline. This serves as direct evidence that BayesNCL does not harm background structures, but rather effectively preserves them while mitigating foreground-background entanglement.
> # 6. [Q2] What is the situation like for other SSLs without negative pairs?
> Thanks for raising the question. Although negative-pair-free SSL is beyond our scope of our paper, this is an interesting question. To this end, we conducted an experiment using VICReg [1] on CIFAR-100, while applying the same non-negativity / Bayesian gating modifications as in NCL/BayesNCL.
>
> | Method                   | Cons. | $H_s$ | $H_m$ | $H_f$ |
> | ------------------------ | ----- | ----- | ----- | ----- |
> | VICReg                   | 1.00  | 4.60  | 4.60  | 4.61  |
> | VICReg + NonNeg          | 1.00  | 4.55  | 4.55  | 4.61  |
> | VICReg + Bayesian Gating | 1.53  | 4.45  | 4.45  | 4.57  |
>
> As shown in the table, VICReg does not exhibit the one-to-one correspondence between semantic factors and feature dimensions, and adding non-negativity or Bayesian gating only yields marginal change. We hope to make it clear that the optimization conflict studied in our paper refers to the per-dimension tug-of-war caused by positive attraction and negative repulsion in contrastive objectives. Since VICReg has no explicit negative pairs, this specific conflict is absent. Its gradients are instead dominated by the invariance term together with variance/covariance regularization, which prevent collapse but do not explicitly encourage semantic factors to align with coordinate axes.
>
> ---
>
> [1] Bardes, A., Ponce, J., & LeCun, Y. (2021). Vicreg: Variance-invariance-covariance regularization for self-supervised learning. _arXiv preprint arXiv:2105.04906_.

---

> > ### Author Rebuttal · Reviewer_VJNk · 2026-04-04
> >
> > Thank you for your rebuttal and providing additional results.
> >
> > - It’s unclear that gradient variance is empirically high. Are there comparable numbers for the authors’ bayesian gated method to show that it improves relatively?
> > - Thank for making the clarifications and edits to address the areas I found confusing.
> > - While BayesNCL does slightly outperform NCL+TopK, it seems slight in both the activations as well as the actual linear performance.
> > - The result using positive alignment SSL methods, in this case, VICReg is interesting. It is does appear that it improves the consistency and other evaluation metrics.
> >
> > Some of my questions have been answered, some are unclear. I will maintain my score.

---

> > > ### Author Response · Authors · 2026-04-04
> > >
> > > Thank you very much for your reply. Please find our response below:
> > > # [Q1]
> > > We have statistically analyzed the same metrics on CIFAR-100 using BayesNCL, and the results are as follows:
> > >
> > > | epoch | AF-GV  | AF-SC   | GV-SC  |
> > > | ----- | ------ | ------- | ------ |
> > > | 10    | 0.337  | -0.954  | -0.268 |
> > > | 20    | 0.285  | -0.919  | -0.189 |
> > > | 30    | 0.178  | -0.893, | -0.070 |
> > > | 40    | -0.066 | -0.887  | 0.177  |
> > > | 50    | 0.326  | -0.866  | -0.189 |
> > > | 60    | 0.469  | -0.836  | -0.339 |
> > > | 70    | 0.519  | -0.858  | -0.398 |
> > > | 80    | 0.644  | -0.869  | -0.468 |
> > > | 90    | -0.008 | -0.789  | -0.001 |
> > > | 100   | 0.563  | -0.863  | -0.399 |
> > >
> > > For ease of comparison, the NCL results are copied below:
> > >
> > > | epoch | AF-GV | AF-SC   | GV-SC  |
> > > | ----- | ----- | ------- | ------ |
> > > | 10    | 0.748 | -0.880  | -0.646 |
> > > | 20    | 0.753 | -0.861  | -0.675 |
> > > | 30    | 0.787 | -0.906  | -0.688 |
> > > | 40    | 0.728 | -0.869  | -0.627 |
> > > | 50    | 0.816 | -0.906, | -0.706 |
> > > | 60    | 0.705 | -0.912  | -0.596 |
> > > | 70    | 0.694 | -0.881  | -0.607 |
> > > | 80    | 0.703 | -0.882  | -0.605 |
> > > | 90    | 0.635 | -0.880  | -0.522 |
> > > | 100   | 0.735 | -0.910  | -0.643 |
> > >
> > > We found that compared to NCL's results, BayesNCL significantly breaks the strong positive correlation between activation frequency and gradient variance, as well as the strong negative correlation between gradient variance and semantics consistency. Furthermore, a noteworthy point is that unlike NCL, BayesNCL's negative correlation between activation frequency and semantic consistency weakens as training epochs increase. We believe these results further prove the correctness of our optimization conflict theory.
> > >
> > > # [Q2, Q4]
> > > We are very happy to see your confusion resolved. Thank you for your careful review.
> > >
> > > # [Q3]
> > > Thank you for your question. However, we would like to clarify that the value of "Act." recorded in Table 1 does not directly indicate the disentanglement effect. We recorded it to show that our method is not simple masking but rather promotes the model to obtain more consistent features. We discussed this metric in the "Interpretability" paragraph of "5.2 Main Results" in the original text. Furthermore, we emphasize that the core contribution of this paper lies in identifying a fundamental conflict of deterministic similarity measures in the feature disentanglement process. Then, starting from the definition of similarity, we derived $S_{IPW}$ and subsequently proposed BayesNCL, which significantly improves the feature disentanglement effect while maintaining or even enhancing performance in downstream tasks.

---

### Decision · Program_Chairs · 2026-04-30

**Decision:**

Accept (regular)

**Comment:**

This paper proposes Bayesian Gated Non-Negative Contrastive Learning (BayesNCL). The main idea for BayesNCL is to address the optimization conflict for NCL methods leading to entangled representations. BayesNCL introduces a probabilistic gating mechanism that dynamically filters out task-irrelevant, high-frequency common features while selectively retaining discriminative semantics. Before the rebuttal this paper has received mixed recommendations: 1 x WR (`VJNk`) 2 x WA (`Q8qy`, `PbsT`), 1 x A (`9W98`).

The reviewers overall appreciated the idea, the theoretical motivation and the solution provided in this paper and the performance gains it achieves.
The main concerns expressed by the reviewers are related to the lack of evidence about gradient oscillations (`VJNk`, `Q8qy`), the writing and organization of the paper (`VJNk`, `Q8qy`,`9W98`), potentially high computational cost (`Q8qy`,`PbsT`), actual harm of learning background features (`VJNk`, `9W98`), potential diminishing utility at scale with large batches (`Q8qy`)

In the rebuttal, the authors provided multiple answers to reviewer concerns including additional experiments and results on gradient oscillation, computational cost, other SSL methods (VICReg), performance with larger batches, sensitivity to learning rate values.


After the rebuttal,`Q8qy`, `PbsT`, `9W98` remained positive, while `VJNk` seems unconvinced by some of the responses and keeps the negative recommendation ad the main message was not clear. The reviewer was not convinced by the fact that BayesNCL improved on specificity of NCL without ultimately improving the representation.
The reviewers discussed their stance and overall remained positive regarding this work.

The meta-reviewer analyzed arguments brought by the reviewers, the paper and the rebuttal. The meta-reviewer agrees with the recommendation of the reviewers and recommends this paper for acceptance.  I encourage the authors to take into consideration the useful advice from the reviewers towards improving this paper for the camera-ready.